# KNOW YOUR BOUNDARIES: THE ADVANTAGE OF EXPLICIT BEHAVIORAL CLONING IN OFFLINE RL

## ABSTRACT

We introduce an offline reinforcement learning (RL) algorithm that explicitly clones a behavior policy to constrain value learning. In offline RL, it is often important to prevent a policy from selecting unobserved actions, since the consequence of these actions cannot be presumed without additional information about the environment. One straightforward way to implement such a constraint is to explicitly model a given data distribution via behavior cloning and directly force a policy not to select uncertain actions. However, many offline RL methods instantiate the constraint indirectly—for example, pessimistic value estimation—due to a concern about errors when modeling a potentially complex behavior policy. In this work, we argue that it is not only viable but beneficial to explicitly model the behavior policy for offline RL because the constraint can be realized in a stable way with the explicitly cloned model. We first suggest a theoretical framework that allows us to incorporate behavior-cloned models into value-based offline RL methods, enjoying the strength of both explicit behavior cloning and value learning. Then, we propose a practical method utilizing a *score-based generative model* for behavior cloning to better handle the complicated behaviors that an offline RL dataset might contain. The proposed method shows state-of-the-art performance on several datasets within the D4RL and Robomimic benchmarks and achieves competitive performance across all datasets tested.

## 1 INTRODUCTION

The goal of offline reinforcement learning (RL) is to learn a policy purely from pre-generated data. This data-driven RL paradigm is promising since it opens up a possibility for RL to be widely applied to many realistic scenarios where large-scale data is available.

Two primary targets need to be considered in designing offline RL algorithms: maximizing reward and staying close to the provided data. Finding a policy that maximizes the accumulated sum of rewards is the main objective in RL, and this can be achieved via learning an optimal Q-value function. However, in the offline setup, it is often infeasible to infer a precise optimal Q-value function due to limited data coverage (Levine et al., 2020; Liu et al., 2020); for example, the value of states not shown in the dataset cannot be estimated without additional assumptions about the environment. This implies that value learning can typically be performed accurately only for the subset of the state (or state-action) space covered by a dataset. Because of this limitation, offline RL algorithms should implement some form of imitation learning objectives that can force a policy to stay close to the given data. Because of this limitation, some form of imitation learning objectives that can force a policy to stay close to the given data warrants consideration in offline RL.

Recently, many offline RL algorithms have been proposed that instantiate an imitation learning objective without explicitly modeling the data distribution of the provided dataset. For instance, one approach applies the pessimism under uncertainty principle in value learning (Buckman et al., 2020; Kumar et al., 2020; Kostrikov et al., 2021a) in order to prevent out-of-distribution actions from being selected. While these methods show promising practical results for certain domains, it has also been reported that such methods fall short compared to simple behavior cloning methods (Mandlekar et al., 2021; Florence et al., 2021) which only model the data distribution without exploiting any reward information. We hypothesize that this deficiency occurs because the imitation learning objective in these methods is only indirectly realized without explicitly modeling the data distribution (e.g. by

pessimistic value prediction). Such an indirect realization could be much more complicated than simple behavior cloning for some data distributions since it is often entangled with unstable training dynamics caused by bootstrapping and function approximation. Hence, implicit methods are prone to over-regularization (Kumar et al., 2021) or failure, and they may require delicate hyperparameter tuning to prevent this deficiency (Emmons et al., 2022). ~~Yet, at the same time, it is obvious that simple behavior cloning cannot extract a good policy from a data distribution composed of suboptimal policies.~~

To this end, we ask the following question in this paper: Can offline RL benefit from explicitly modeling the data distribution via behavior cloning no matter what kind of data distribution is given? Previously, there have been attempts to use an explicitly trained behavior cloning model in offline RL (Wu et al., 2019; Kumar et al., 2019; Fujimoto et al., 2019; Liu et al., 2020), but we argue that two important elements are missing from existing algorithms. ~~First, high-fidelity behavior cloning has not been achieved, despite the need in offline RL for precise estimation of behavior policy~~ (Levine et al., 2020). First, high-fidelity generative models have not been integrated with offline RL algorithms even though inaccurate estimation of behavior policy could limit the final performance of the algorithm (Levine et al., 2020). Florence et al. (2021) have tried an energy-based generative model, but the proposed method is an imitation learning that does not incorporate a value function. ~~Second, the trained behavior cloning models have only been utilized with heuristics or proxy formulations that are only empirically justified~~ (Wu et al., 2019; Kumar et al., 2019). Second, the trained behavior cloning models have only been utilized with certain limited forms, such as KL (Wu et al., 2019) or MMD (Kumar et al., 2019) divergence between the cloned policy and an actor policy. Therefore, we tackle these two problems by: first, incorporating a state-of-the-art score-based generative model (Song & Ermon, 2019; 2020; Song et al., 2021) to fulfill the high-fidelity required for offline RL, and second, by proposing a theoretical framework, direct Q-penalization (DQP), that provides a mechanism to integrate the trained behavior model into value learning. Furthermore, DQP can provide an integrated view of different offline RL algorithms, helping to analyze the possible failures of these algorithms.

We evaluate our algorithm on various benchmark datasets that differ in quality and complexity, namely D4RL and Robomimic. Our method shows not only competitive performance across different types of datasets but also state-of-the-art results on complex contact-rich tasks, such as the transport and tool-hang tasks in Robomimic. The results demonstrate the effectiveness of the proposed algorithm as well as the practical advantage of explicit behavior cloning, which was previously considered a bottleneck that would limit the final offline RL performance (Levine et al., 2020) ~~unnecessary or infeasible~~.

To summarize, our contributions are: (1) We provide a theoretical framework for offline RL, DQP, which provides a unified view of previously disparate offline RL algorithms; (2) Using DQP, we suggest a principled offline RL formulation that incorporates an explicitly trained behavior cloning model; (3) We propose a practical algorithm that instantiates the above formulation, leveraging a score-based generative model; and (4) we achieve competitive and state-of-the-art performance across a variety of offline RL datasets.

## 2 RELATED WORKS

The end goal of offline RL is to extract the best possible policy from a given dataset, regardless of the quality of the trajectories that compose the dataset (Ernst et al., 2005; Riedmiller, 2005; Lange et al., 2012; Levine et al., 2020). One of the simplest approaches to tackle this problem is imitation learning (IL) (Schaal, 1999; Florence et al., 2021) hoping to recover the performance of the behavior policy which generated the dataset. However, simple imitation would fail to achieve the end goal of offline RL since one cannot outperform the behavior policy by just imitating it.

This problem is commonly addressed with value learning, trying to resolve the distribution shift problem that arises in the offline setup. Since distribution shift commonly results in overestimation of values, offline RL algorithms try to estimate values pessimistically for out-of-distribution inputs (Kumar et al., 2020; Goo & Niekum, 2021), sometimes by explicitly quantifying the certainty with a trained transition dynamics model (Yu et al., 2020; Kidambi et al., 2020), a generative model (Rezaeifar et al., 2021), or a pseudometric (Dadashi et al., 2021). The distribution shift is also commonly addressed by constraining a policy to be close to the behavior one. Specifically, based on

how the constraint is instantiated, policy-constraint methods can be further categorized into implicit methods, which constrain a policy via weighted behavior cloning or linear gradient combination (Peng et al., 2019; Wang et al., 2020; Kostrikov et al., 2021b; Brandfonbrener et al., 2021; Fujimoto & Gu, 2021; Wang et al., 2022), and explicit methods (Fujimoto et al., 2019; Kumar et al., 2019; Wu et al., 2019; Liu et al., 2020), which constrain the policy learning via value penalty or policy regularization.

We generally follow the learning structure of the explicit policy-constraint method. Especially, our offline RL algorithm is closely related to the works of Fujimoto et al. (2019); Wu et al. (2019); Liu et al. (2020) in which a behavior policy $\beta$ is explicitly cloned first, and the cloned policy is directly used to instantiate the policy-constraint. However, we try to enhance its performance by first, suggesting a principled policy-constraining method that utilizes the cloned policy, and second, using a powerful generative model for explicit behavior cloning that can greatly reduce the BC error and thereby allow precise policy regularization.

Concurrent with our work, Wang et al. (2022) propose an offline RL algorithm that leverages a diffusion-based generative model. Their method is based on the works of Fujimoto & Gu (2021) in which the ordinary actor loss is linearly combined with behavior cloning loss. The proposed algorithm is simple and minimalistic, but the theoretical justification of the approach has not been fully addressed.

## 3 PRELIMINARIES

We use a Markov Decision Process (MDP) as a foundation for our mathematical framework. An MDP is defined by a tuple $\mathcal{M} = (\mathcal{S}, \mathcal{A}, T, d_0, r, \gamma)$; a set of states $s \in \mathcal{S}$, a set of actions $a \in \mathcal{A}$, transition dynamics $T = p(s'|s, a)$, an initial state distribution $d_0(s_0)$, a reward function $r(s, a)$, and a discount factor $\gamma$. In this setup, the goal of reinforcement learning is to find an optimal policy $\pi^*(a|s)$ that maximizes the expected sum of discounted reward (return) $J(\pi)$:

$$\pi^* = \arg\max_\pi J(\pi) = \arg\max_\pi \mathbb{E}_{\tau \sim \rho_\pi} \left[ \sum_{t=0}^{H} \gamma^t r(s_t, a_t) \right], \tag{1}$$

where $\tau$ is a sequence of states and actions $(s_0, a_0, \cdots, s_H, a_H)$ of length $H$, and $\rho_\pi$ is a trajectory distribution of a policy $\pi$, which can be represented as $\rho_\pi(\tau) = d_0(s_0) \prod_{t=0}^{H} \pi(a_t|s_t) T(s_{t+1}|s_t, a_t)$.

We can directly optimize the return when we can compute the gradients of $J(\pi_\phi)$ with respect to the policy parameters $\phi$ (Williams, 1992; Schulman et al., 2017; 2015), but this approach is not ~~straightforwardly~~ extend to an offline setting since on-policy data is typically required to compute the gradient. Instead, it is more common for offline RL methods to extend dynamic programming approaches which are formed around the action value function $Q^\pi(s, a)$ which is formally defined as: $Q^\pi(s_t, a_t) = \mathbb{E}_{\tau \sim \rho_\pi(s_t, a_t)} \left[ \sum_{t'=t}^{H} \gamma^{t'-t} r(s_{t'}, a_{t'}) \right]$. The Q function of a certain policy $\pi_k$ always implies a greedy policy $\pi_{k+1}$, which is better than or equal to the evaluation target policy $\pi_k$. Therefore, an optimal policy can be found by iteratively evaluating a Q function for a new greedy policy $\pi_{k+1}$ until convergence:

$$\begin{cases} Q^{\pi_k} = \lim_{n \to \infty} \left( \mathcal{B}^{\pi_k} \right)^n Q^{\pi_{k-1}} & \text{(policy evaluation)}, \\ \pi_{k+1}(a|s) = \arg\max_a Q^{\pi_k}(s, a) & \text{(policy improvement)}, \end{cases} \tag{2}$$

where $\mathcal{B}^\pi$ is the Bellman operator, which has the ground truth $Q^\pi$ as a unique fixed point (Lagoudakis & Parr, 2003). This algorithm is called policy iteration. Although the convergence of the algorithm is restricted to the scenario where the unique fixed point is reachable (Sutton & Barto, 2018), policy iteration has been widely used as a backbone for most offline RL algorithms due to its extensibility to off-policy data; policy evaluation can be done with off-policy data using bootstrapping. When the value function and the policy are represented with parameters $\theta$ and $\phi$ respectively, the policy iteration algorithm with bootstrapping has the following form:

$$\begin{cases} \theta_{k+1} \leftarrow \arg\min_\theta \mathbb{E}_{s,a,r,s' \sim D} \left[ d\big(Q_\theta(s, a), r + \gamma \mathbb{E}_{a' \sim \pi_k(a'|s')} Q_k(s', a')\big) \right] \\ \phi_{k+1} \leftarrow \arg\max_\phi \mathbb{E}_{s \sim D, a \sim \pi_\phi(a|s)} \left[ Q_{k+1}(s, a) \right] \end{cases} \tag{3}$$

where $k$ is an update step, $d$ is a distance metric such as squared $l_2$ or Huber loss, and $D$ is a provided (offline) dataset that contains transition tuples $D = (s, a, r, s')$. For the brevity of notation, we denote $Q_{\theta_k} := Q_k$ and $\pi_{\phi_k} := \pi_k$.

## 4 METHOD

In this paper, we consider offline RL algorithms that utilize the policy iteration scheme shown in Eq. 3, which covers several offline RL methods (Kumar et al., 2020; 2019; Wu et al., 2019). In this family of methods, the correctness of the value function becomes the major concern since the policy evaluation could diverge due to the data restriction imposed by the offline setup. Specifically, divergence can happen because of the bootstrapping and the function approximation; when some estimates are erroneously high due to poor generalization, the over-estimated values are likely to be picked up on the policy improvement step and feed back to policy evaluation via bootstrapping, completing a vicious cycle that causes training to diverge. Therefore, offline RL methods focus on solving the overestimation problem with different regularization methods, such as policy constraints (Kumar et al., 2019; Wu et al., 2019) or pessimism (Kumar et al., 2020; Goo & Niekum, 2021).

One straightforward solution to the over-estimation problem is directly penalizing Q estimation (Rezaeifar et al., 2021; Dadashi et al., 2021) with a penalty function $p(s, a)$: $\tilde{Q}_\theta(s, a) = Q_\theta(s, a) - p(s, a)$. This can be a solution because this penalty function, if chosen carefully, can reduce erroneously high values and prevent them from being propagated via bootstrapping. We refer to this family of algorithms as Direct Q penalization (DQP).

In DQP, we can easily observe that the penalty function that describes the oracle Q estimation error (i.e. $p(s, a) = Q_\theta - Q^{\pi_\phi}$) is the best solution. Therefore, we want to design a penalty function that resembles the oracle error. Since the estimation error is likely to occur more often for out-of-distribution state-action pairs $(s, a') \notin D$, a few ~~ad hoc~~ methods have been proposed for the purpose of measuring the eestimation error or epistemic uncertainty of $Q_\theta$; the aleatoric uncertainty of the transition dynamics model is suggested as a proxy for the epistemic uncertainty of $Q_\theta$ (Yu et al., 2020), and generative models (Rezaeifar et al., 2021) or pseudometrics (Dadashi et al., 2021) are proposed with the purpose of distinguishing whether a particular $(s, a)$ is in-distribution or not.

~~However, it has not been thoroughly investigated how penalty functions affect the policy iteration process, nor what penalty functions are best for offline RL. Without answers to these questions, DQP methods can only be understood as ad-hoc methods in which heuristically designed penalty functions are used to prevent the overestimation.~~ DQP provides a unified way to represent different offline RL algorithms in terms of penalty functions. However, in order for a unified perspective to provide a better understanding of offline RL algorithms and thereby be useful for developing better offline RL algorithms, it is necessary to investigate the effect of the penalty function under the policy iteration scheme. To this end, we address the following questions: (1) What is the effect of direct Q-penalization in the context of the policy iteration framework?; (2) How can we design an appropriate penalty function based on this analysis?; (3) How can we instantiate the penalty function and achieve strong performance across different offline RL datasets?

### 4.1 THEORETIC BACKGROUND ON DIRECT Q-PENALIZATION

We describe a theorem that answers the first question: soft policy iteration (Haarnoja et al., 2018) with a penalized value function $\tilde{Q}$ is equivalent to policy iteration regularized by $D_{\mathrm{KL}}\big(\pi(s)\|\pi_p(s)\big)$ where $\pi_p(a|s) := \mathrm{softmax}\big(-p(s, a)\big)$. This theorem is a generalized version of the theorem shown in (Rezaeifar et al., 2021), which does not require unnecessary assumptions on the penalty function.

**Theorem 1** (Equivalence between KL-policy regularization and DQP ). *The following two algorithms are equivalent.*
Policy iteration w/ KL-policy regularization:

$$\begin{cases} \theta_{k+1} \leftarrow \arg\min_\theta \mathbb{E}_{s,a,r,s'\sim D}\Big[d\Big(Q(s,a), r + \gamma\langle\pi_k, Q_k\rangle(s') - \gamma D_{\mathrm{KL}}\big(\pi_k(s')\|\pi_p(s')\big)\Big)\Big] \\ \phi_{k+1} \leftarrow \arg\max_\phi \mathbb{E}_{s\sim D}\Big[\langle\pi_\phi, Q_{k+1}\rangle(s) - D_{\mathrm{KL}}\big(\pi_\phi(s)\|\pi_p(s)\big)\Big]. \end{cases}$$

Soft policy iteration (Haarnoja et al., 2018) w/ penalty:

$$\begin{cases} \theta_{k+1} \leftarrow \arg\min_\theta \mathbb{E}_{s,a,r,s'\sim D} \left[ d\Big( Q(s,a), r + \gamma\big( \langle \pi_k, Q_k - p \rangle(s') - Z(s') + \mathrm{H}\big(\pi_k(s')\big) \big) \Big) \right] \\ \phi_{k+1} \leftarrow \arg\max_\phi \mathbb{E}_{s\sim D} \left[ \langle \pi_\phi, Q_{k+1} - p \rangle(s) + \mathrm{H}\big(\pi_\phi(s)\big) \right] \end{cases}$$

where $d$ is a distance metric, $\langle u_1, u_2 \rangle := \sum_a u_1(\cdot, a) u_2(\cdot, a)$, and $Z(s) = \ln \sum_a \exp\big( -p(s,a) \big)$.

*Proof.* The common term in KL policy regularization can be rearranged as follows:

$$\begin{aligned} \langle \pi, Q \rangle(s) - D_{\mathrm{KL}}\big(\pi(s)\|\pi_p(s)\big) &= \langle \pi, Q \rangle(s) - \langle \pi, \ln\pi - \ln\pi_p \rangle(s) \\ &= \langle \pi, Q + \ln\pi_p \rangle(s) - \langle \pi, \ln\pi \rangle(s) \\ &= \langle \pi, Q - p \rangle(s) - Z(s) + \mathrm{H}\big(\pi(s)\big). \end{aligned}$$

Then, we can get the equivalence when we replace the term with the rearranged term. Note that the normalization term $Z(s)$ can be dropped in the policy update step since $Z(s)$ is not a function of $\phi$. Also, we can safely ignore $Z(s)$ in the policy evaluation step when $|Z(s) - Z(s')| < \epsilon$ for any pair of $(s, s') \in \mathcal{S} \times \mathcal{S}$, because it does not affect the policy improvement step. □

The theorem is straightforward since it describes a special case of regularized policy iteration Geist et al. (2019), namely KL-control (Peters et al., 2010; Schulman et al., 2015), in which the policy is regularized through KL-divergence with respect to another policy. Yet, the theorem shows a way to interpret any penalty function from the point of view of KL-control and vice versa. Therefore, by using the theorem, we can have a unified view of previously disparate offline RL algorithms. In Table 1, we compare different offline RL algorithms in terms of the penalty function that each algorithm uses.

## 4.2 WHAT MAKES A GOOD PENALTY FUNCTION?

Theorem 1 shows the connection between a penalty function and its effect as a policy regularizer, and can help to guide the construction of an effective, principled penalty function. Specifically, we propose a penalty function that can instantiate the support set constraint (Kumar et al., 2019; Liu et al., 2020), which restricts the action space of a trained policy to be in the support set of the behavior policy $\beta(a|s)$. The support set constraint is an effective way to solve the offline RL problem in that the suboptimality caused by the constraint is bounded (Kumar et al., 2019; Liu et al., 2020). While previous works express the constraint in terms of the distribution-constrained Bellman operator, we represent the constraint via a penalty function since it allows us to compare different offline RL algorithms under the same viewpoint, helping to analyze the possible failures of these algorithms.

The following penalty function instantiates the support set constraint:

$$p(s,a) = \begin{cases} 0 \text{ for } \{(s,a)|\beta(a|s) \geq \epsilon\} \\ \infty \text{ otherwise} \end{cases} \tag{4}$$

where $\epsilon$ is a threshold hyperparameter to decide whether $(s, a)$ is considered out-of-support or not. The penalty function carries the same effect as the filtration operator in (Liu et al., 2020) under the policy iteration scheme. This is because the function prevents out-of-support actions from being chosen by the policy while it imposes no preference over in-support actions; a rare action that has not occurred often in a dataset can be selected as long as it provides a high Q value. This indifference is a desirable property when good trajectories compose only a small portion of a dataset, since good actions could be drowned out by more frequent actions if the penalty function is designed to prefer more frequent actions.

We can also confirm the characteristics of the penalty function by observing the flip side: the penalty-induced policy $\pi_p$ and the KL-constraint $D_{\mathrm{KL}}(\pi\|\pi_p)$. Since the reverse KL term makes $\pi$ seek a mode of $\pi_p$ which is the uniform distribution for the in-support actions, the policy $\pi$ is guided to select one of the actions in the support set while there is no preference over actions in the set. Therefore, the penalty function instantiates the support set constraint.

Given the proposed penalty function, the similarity between different offline RL methods can be observed. For instance, we can see that BRAC-KL and CQL penalize the out-of-support actions infinitely: when $\beta(a|s)$ is zero, the penalty becomes infinite. However, some discrepancies can also be noted, and this provides some hints on how and why other methods could fail due to excessive or insufficient pessimism.

First, BRAC-KL could fail because it prefers actions that are more frequently executed by the behavior policy. This can be easily seen from the KL-policy regularization perspective. Since $\pi_p(s) = \beta$ in BRAC-KL, the penalty would make a policy to seek the mode of $\beta(a|s)$ when the KL regularization term dominates the policy update. Therefore, the algorithm could work like behavior cloning that disregards rare but good actions in the provided dataset. CQL could also exhibit a similar problem since the penalty function is defined with $\beta$. Like BRAC-KL, when the value function implied policy $\mu_k$ selects an action that is infrequent in $\beta$, it will be harshly penalized. Especially when CQL is tuned to strongly penalize the out-of-support action (i.e., $\alpha_k$ is large), it could force the policy to mimic the dataset (Kumar et al., 2021). Similarly, TD3+BC and Diffusion RL (Wang et al., 2022) could suffer from the same problem when the penalty strength $\alpha$ is not properly tuned.

Another common problem that arises in other penalty functions is their use of proxies and their formulation that replace the $\beta(a|s)$; for example, a conditional variational autoencoder (CVAE)(Rezaeifar et al., 2021), a pseudometric (Dadashi et al., 2021), or a transition dynamics model (Yu et al., 2020) are estimated instead of the behavior policy $\beta$, and penalty functions are designed heuristically with the proxy estimates. While such formulations could show some positive correlation to the suggested penalty function, there is no clear connection that allows us to interpret the penalty in terms of $\beta$ or the support set.

BEAR, on the other hand, is designed to implement the support set constraint as ours, so it could avoid the problem of BRAC-KL or CQL that prefers more frequent actions. However, BEAR could be inaccurate since they instantiate the constraint without explicitly modeling the behavior policy $\beta$. Especially, they resort to maximum mean discrepancy (MMD) distance since it can be computed only using the samples from a dataset, but the use of the distance metric is only empirically justified (Kumar et al., 2019). In contrast, we directly instantiate the support set constraint by explicitly modeling the behavior policy.

### 4.3 PRACTICAL IMPLEMENTATION

We now propose a practical algorithm that instantiates the penalty function designed above. Essentially, the designed penalty function serves to determine whether an action $a$ at a certain state $s$ is likely to be executed by the behavior policy $\beta$. Therefore, we can implement the penalty function simply by cloning the behavior policy explicitly and checking the likelihood $\beta(a|s)$ with the cloned model.

There have been other research works that have tried to model a behavior policy using generative models, such as variational auto-encoders (VAEs) (Fujimoto et al., 2019; Rezaeifar et al., 2021). However, the performance of these approaches is limited compared to methods that do not explicitly clone the behavior policy $\beta$. We presume that the reason for this failure is the limited expressivity of the generative model; since the behavior policy $\beta$ can be complex, discontinuous, and multi-modal, only a very expressive model can successfully model the policy. To this end, we chose to use a

Table 1: The penalty functions of different offline RL algorithms.

| | $p(s, a)$ | Remark |
|---|---|---|
| BRAC-KL | $-\log \beta(a\|s)$ | |
| BRAC-MMD$^2$ | $\mathrm{MMD}^2(\pi_k, \beta)$ | |
| TD3+BC | $-\alpha(a - \beta(s))^2$ | Wang et al. (2022) is also similar. |
| Anti-Exploration | $\alpha\|a - \mathrm{Dec}(\mathrm{Enc}(s,a))\|_2^2$ | Enc and Dec are conditional VAE. |
| PLOFF | $\alpha_1 Q_k(s,a)\exp(-\alpha_2 \mathrm{D}(s,a))$ | D is pseudometric. |
| MOPO | $\alpha\|\Sigma(s,a)\|$ | $\Sigma$ is the std. of the trained $\mathcal{T}$. |
| CQL | $\alpha_k[\frac{\mu_k}{\beta} - 1]$ | $\mu_k$ is the soft-policy given $Q_k$. |

score-based generative model (Song & Ermon, 2019; 2020; Song et al., 2021), which has recently shown great success in generating high-quality images. Furthermore, the score-based generative model allows an exact likelihood computation which is essential in instantiating the penalty function. We briefly examine the ability of the score-based generative model using four discontinuous multi-modal distributions, and the results are shown in Figure A.1. In all four cases, the inferred probability distribution is very sharp, and its log probability resembles the penalty function we proposed.

In the score-based generative model, a target distribution $p(x)$ is indirectly expressed and trained in the form of the gradient of a log probability density function $\nabla_x \log p(x)$, often referred to as the (Stein) score function (Liu et al., 2016). When we approximate the score function of a behavior policy $\beta(a|s)$ accurately via score-matching algorithm (i.e. $s_\psi(a|s) \approx \nabla_a \beta(a|s)$) (Song & Ermon, 2019), we can instantiate the penalty function with a hyperparameter $\epsilon$ which decides whether a certain action $a$ given $s$ is considered to be sufficiently in-support or not. While this formulation allows direct instantiation of the penalty function that can be plugged into the DQP framework, it is computationally prohibitive. This is because, first, we need to run an iterative algorithm to compute the log-likelihood from the score function, and second, it could hurt the generalization performance of the value function since Q has to output an extremely wide range of values including negative infinity. Therefore, we propose a practical approximation of the policy iteration algorithm that utilizes the proposed penalty function.

The key observation is that the policy trained on top of the penalized value function will never select out-of-support actions due to penalization. This allows two modifications to the original policy iteration algorithm: First, we can perform policy evaluation only considering in-support state-action pairs; i.e., we can bootstrap from one of the samples from $\beta(a|s)$, and we can skip the penalty computation since the penalty is zero for in-support data-points under the suggested penalty function. While sampling using the score function also requires expensive iterative computation, we can greatly reduce the computation by prepopulating samples for states that exist in the dataset and repeatedly using it in the policy evaluation step. Specifically, the policy evaluation is done with the following loss function:

$$L_{\text{policy-eval}} = \mathbb{E}_{s,a,r,s'\sim D}\left[d\left(Q_\theta(s,a), r + \gamma \underset{a'\in\widehat{\text{supp}}(\beta(s'))}{K\text{-th}} \left[Q_{\tilde\theta}(s',a')\right]\right)\right] \tag{5}$$

where $Q_{\tilde\theta}$ is a slowly updated target network, $\widehat{\text{supp}}(\beta)$ is a set of samples approximating $\text{supp}(\beta)$, and $K$-th is an operator that selects the $K$-th item among candidates. When $K = 1$, it becomes $\max$ operator. Both $Q_{\tilde\theta}$ and $K$-th operators are adapted to stabilize the learning.

Second, we can skip the policy improvement step since policy evaluation is done with pre-generated samples, not depending on any parameterized policy. Instead, we can define an implicit policy using the last $Q_\theta$ and $s_\psi$:

$$\pi(a|s) = \frac{\exp\left(\alpha Q_\theta(s,a)\right)}{\sum_{a'\in\widehat{\text{supp}}(\beta)} \exp\left(\alpha Q_\theta(s,a')\right)} \quad \text{or} \quad \frac{\exp\left(\alpha A(s,a)\right)}{\sum_{a'\in\widehat{\text{supp}}(\beta)} \exp\left(\alpha A(s,a')\right)} \tag{6}$$

where $A(s,a) = Q_\theta(s,a) - \frac{1}{|\widehat{\text{supp}}(\beta)|}\sum_{a'\in\widehat{\text{supp}}(\beta)} Q_\theta(s,a')$ is an advantage function, and $\alpha$ is a temperature parameter that controls the policy softness with regard to the $Q$ or $A$; when $\alpha$ is zero or infinity, the resulting policy becomes $\beta(a|s)$ or greedy with regard to $Q_\theta$, respectively. To sample an action from the policy $\pi$, we can sample one action from an empirical action distribution consisting of samples generated from $\beta$ on the fly. Alternatively, we can also train parameterized policy $\pi_\phi(a|s)$ using advantage-weighted regression (AWR) (Neumann & Peters, 2008; Peng et al., 2019) with the advantage function $A(s,a)$.

The resulting algorithm can be regarded as one special type of Q-learning in which we restrict the domain of the maximum operator to in-support actions. We refer to this algorithm as Action-Restricted Q-learning (ARQ). The pseudocode of ARQ is shown in Algorithm 1. Also, ARQ can be seen as an extension of MBS-QI (Liu et al., 2020) in that it makes the existing algorithm applicable to MDP with a continuous action space. Note that ARQ is one way to instantiate the penalty function, mainly due to the expensive computational cost of the score-based generative model. We discuss about other possible instantiations in the discussion section.

---

**Algorithm 1:** Action-Restricted Q-learning (ARQ)

---

**Input :** Dataset $D = \{(s, a, r, s')\}$, Hyperparameter $N, \epsilon, K, \alpha$
Initialize $s_\psi(a|s)$, $Q_\theta(s, a)$, and $\pi_\phi(a|s)$ (if needed)
Train $s_\psi$ with a score matching algorithm (Song et al., 2021)
Sample $N$ in-support actions for $s \in D$ (i.e., $\beta_\psi(a|s) > \epsilon$)
**while** *until convergence* **do**
   | Update $\theta$ with $\nabla_\theta L_{\text{policy-eval}}$ (Eq. 5)
**while** *until convergence* **do**
   | Update $\phi$ with $\nabla_\phi - \mathbb{E}_{s,a \sim D}\left[e^{\alpha A(s,a)} \log \pi_\phi(a|s)\right]$; AWR
**return** $s_\psi$, $Q_\theta$, and $\pi_\phi$

---

## 5 EXPERIMENTS

Our empirical goal is to design an algorithm that enjoys the strength of both explicit behavior cloning and value learning. Therefore, the main goal of the experiments is to check whether the proposed algorithm ARQ achieves competitive performance on different types of datasets, ranging from a dataset that consists of near-optimal data in which explicitly cloning a behavior is adequate, to a dataset containing various suboptimal trajectories in which learning a value function is necessary.

The implementation of ARQ consists of four steps: score-based generative model $s_\psi$ learning, sampling, Q-learning $Q_\theta$, and optional explicit policy $\pi_\phi$ training. As for the hyperparameters, we tune the hyperparameters $K$ and $\alpha$ for each group of datasets using random search while we use $N = 30$ and $\epsilon = e^{-5}$ (i.e., 30 samples are generated and dropped if the likelihood is lower than $\epsilon = e^{-5}$) all across the datasets tested. For the detailed implementation details and hyperparameters, please refer Appendix A.2 or the provided code [1].

The proposed method is evaluated on various simulated benchmark datasets from simple low-dimensional locomotion tasks to complex contact-rich manipulation tasks. Specifically, we use locomotion (Brockman et al., 2016), Adroit (Rajeswaran et al., 2018), Kitchen (Gupta et al., 2019), and Antmaze tasks in D4RL (Fu et al., 2020), and six manipulation tasks in Robomimic (Mandlekar et al., 2021). We use medium-replay, medium, expert, and medium-expert datasets of the locomotion task. We use machine-generated (mg.), proficient-human (ph.), and multi-human (mh.) datasets of Robomimic, each of which consists of a replay buffer of an SAC training run, trajectories of a proficient human demonstrator, and trajectories of multiple human demonstrators with different levels of proficiency.

We compare the proposed method to behavior cloning baselines, specifically ordinary BC and implicit behavior cloning (Florence et al., 2021), and prior state-of-the-art offline RL methods. Namely, we compare the performance of our method with TD3+BC (Fujimoto & Gu, 2021), Decision Transformer (DT) (Chen et al., 2021), One-step RL (Brandfonbrener et al., 2021), CQL (Kumar et al., 2020), and IQL (Kostrikov et al., 2021b). The aggregated results are displayed in Table 2.

The proposed algorithm ARQ shows competitive performance on all ranges of datasets, from near-optimal ones in which simple behavior cloning is sufficient, to suboptimal datasets in which value learning is necessary. Also, ARQ exhibits state-of-the-art performance on complex and contact-rich tasks, such as adroit, kitchen, and Robomimic datasets. The results indicate the practical effectiveness of the proposed algorithm, as well as the advantage of performing behavior cloning explicitly with high-fidelity models.

To examine the importance of each component in ARQ, we run two ablations; we evaluate an implicit policy defined only with $s_\psi$ without any value function, and an implicit policy incorporating $Q^\beta$ instead of ARQ. The results of the ablations affirm our hypotheses. First, when the dataset is near-optimal (e.g., adroit-human or proficient-human datasets), explicitly modeling the behavior policy can address the problem, and similar performance is obtained when we use ARQ. Next, we confirm the necessity of value learning and the ability of ARQ in leveraging the explicitly cloned behavior model in learning a value function. Especially in the tasks where trajectory stitching is required (e.g., kitchen and antmaze datasets), we can see performance improvement from $s_\psi$ and $Q(\beta) + s_\psi$ to ARQ+$s_\psi$, and we achieve state-of-the-art performance with the help of the explicit models.

---

[1] Please see the attached supplementary files. The code will be disclosed to public upon publication.

Table 2: Aggregated performance of prior methods, ours, and two ablations ($s_\psi$ and $Q(\beta) + s_\psi$) on D4RL (Fu et al., 2020) and Robomimic (Mandlekar et al., 2021) datasets. Each number represents the mean relative performance over 100 episodes. 0 and 100 represent the performance of random and expert policy, respectively. Unless noted as (ours) or (repro.), all the numbers are borrowed from Kostrikov et al. (2021b); Fujimoto & Gu (2021); Mandlekar et al. (2021). The numbers generated by us are averaged over 3 different random seeds. We run IQL on Robomimic by ourselves using the author-provided implementation.

| | Without reward | | | With $Q^\beta$ | | | With reward / value function | | | | |
|---|---|---|---|---|---|---|---|---|---|---|---|
| | BC | Impl. BC | $s_\psi$ (ours) | One-step | $Q(\beta)$ +$s_\psi$ (ours) | DT | TD3 +BC | CQL | IQL | ARQ +$\pi_\phi$ (ours) | ARQ +$s_\psi$ (ours) |
| **locomtion-v2 (total)** | 739 | 521 | 639 | | 911 | | **992** | 996 | **1007** | **1000** | 947 |
| **adroit-v0 (total)** | 105 | | 116 | | 160 | | | 94 | 118 | 95 | **161** |
| human-v0 (total) | 67 | **99** | 87 | - | 89 | - | - | 52 | 77 | 45 | 90 |
| cloned-v0 (total) | 37 | - | 29 | 62 | **71** | - | - | 42 | 41 | 50 | **71** |
| **kitchen-v0 (total)** | 155 | 160 | 170 | | 186 | | | 145 | 160 | 126 | **204** |
| complete | 65 | **85** | 74 | - | 75 | - | - | 44 | 63 | 37 | 77 |
| partial | 38 | 38 | 45 | - | 59 | - | - | 50 | 46 | 50 | **70** |
| mixed | 52 | 38 | 51 | - | 52 | - | - | 51 | 51 | 39 | **57** |
| **antmaze-v0 (total)** | 100 | | 121 | 125 | 215 | 112 | 164 | 304 | 378 | **416** | 327 |
| umaze | 55 | - | 58 | 64 | 81 | 59 | 79 | 74 | 88 | **97** | 94 |
| umaze-div. | 46 | - | 61 | 61 | 62 | 53 | 71 | **84** | 62 | 62 | 58 |
| med.-play | 0 | - | 1 | 0 | 25 | 0 | 11 | 61 | 71 | **80** | 69 |
| med.-div. | 0 | - | 1 | 0 | 45 | 0 | 3 | 54 | 70 | **82** | 65 |
| large-play | 0 | - | 0 | 0 | 1 | 0 | 0 | 16 | **40** | 37 | 18 |
| large-div. | 0 | - | 0 | 0 | 1 | 0 | 0 | 15 | 48 | **58** | 23 |
| **D4RL (total)** | 1,099 | | 1,046 | | 1,472 | | | 1,538 | **1,663** | 1,637 | 1,639 |

| | BC★ | $s_\psi$ (ours) | $Q(\beta)$ +$s_\psi$ (ours) | BCQ★ | CQL★ | IQL (repro.) | ARQ +$\pi_\phi$ (ours) | ARQ +$s_\psi$ (ours) |
|---|---|---|---|---|---|---|---|---|
| **robomimic (total)** | 701 | 644 | **749** | 592 | 281 | 342 | 659 | **750** |
| mg.-lift | 65 | 29 | 86 | **91** | 64 | 79 | 79 | 82 |
| mg.-can | 65 | 19 | 55 | **75** | 1 | 62 | **76** | 60 |
| ph.-lift | **100** | 99 | **100** | **100** | 93 | 58 | **100** | 98 |
| ph.-can | **95** | 95 | 93 | 89 | 38 | 26 | 92 | 95 |
| ph.-square | **79** | 66 | 72 | 50 | 5 | 24 | 44 | 69 |
| ph.-transport | 17 | **27** | **28** | 7 | 0 | 1 | **29** | **30** |
| ph.-toolhang | 29 | **70** | 64 | 0 | 0 | 3 | 3 | **71** |
| mh.-lift | **100** | 96 | 94 | **100** | 57 | 51 | 99 | 95 |
| mh.-can | **86** | 84 | 89 | 63 | 22 | 25 | 90 | 86 |
| mh.-square | 53 | 44 | 51 | 14 | 1 | 12 | 31 | 51 |
| mh.-transport | 11 | **15** | **17** | 3 | 0 | 0 | **16** | 13 |
| **D4RL + robomimic** | 1,799 | 1,690 | 2,221 | | 1,818 | 2,005 | 2,296 | **2,389** |

★ represents that the best performance during training iterations is picked a posteriori.

It is also noteworthy that the ablated method with $Q^\beta$ shows competitive results on a large number of benchmarks. Echoing prior research (Goo & Niekum, 2021; Brandfonbrener et al., 2021), these results indicate that the vast majority of offline RL benchmarks can be resolved without iterative value learning, while most offline RL algorithms tackle problems that arise from it. Therefore, in order to fairly evaluate the offline RL algorithms and thereby foster the advance of the offline RL field, it is essential to focus on environments that require value learning (e.g., antmaze) or develop new benchmarks.

## 6 DISCUSSION

We investigate an offline RL algorithm that combines explicit behavior cloning and value learning. We provide a theoretical framework, DQP, which enables various offline RL algorithms to be expressed in terms of different penalty functions, and we derive a principled penalty function that can leverage a behavior cloning model. Then, we provide a practical algorithm, ARQ, which realizes the derived

penalty function. We implement the algorithm with a powerful generative model to maximize the full potential of ARQ. As a result, the proposed algorithm shows competitive results on most of the D4RL and Robomimic benchmarks and yields state-of-the-art results in several tasks. ~~This indicates that the common presumption—that it is unnecessary or infeasible to estimate a behavior policy in offline RL—is likely incorrect.~~ These results indicate that explicitly cloning a behavior policy can be actually advantageous, which has been avoided because of the performance limitations that can arise from inaccurate modeling of the policy.

The major drawback of the proposed algorithm is the computationally expensive sampling procedure. While it needs to be computed only once before the value learning step, it can take several hours to generate samples (90K samples are generated per hour on our in-house workstation with an Nvidia GTX 1080 Ti). Therefore, future research may examine how to reduce computational burden, for instance, by using different generative models for behavior cloning. Or, it may be possible to devise a method that directly utilizes the score function under the actor-critic framework; since the actor update step with the penalized Q function ($\tilde{Q} = Q - p$) only requires a gradient of $p$, not an exact penalty value, the computational bottleneck may be avoided if the gradient of the penalty function can be computed directly from the score function.

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

# A APPENDIX

## A.1 SCORE-BASED GENERATIVE MODEL (SONG & ERMON, 2019; 2020; SONG ET AL., 2021)

In the score-based generative model, a target distribution $p(x)$ is indirectly expressed and trained in the form of the gradient of a log probability density function $\nabla_x \log p(x)$, often referred to as the (Stein) score function (Liu et al., 2016). This method circumvents several problems of other generative models. The main advantages are, first, we can circumvent the problem of the inferring normalizing constant that arises in likelihood-based methods (Kingma & Welling, 2013; LeCun et al., 2006), and second, we can train the score function without worrying about training instability that arises in adversarial training Goodfellow et al. (2014) via score-matching algorithms Hyvärinen & Dayan (2005); Vincent (2011); the score-matching algorithm minimizes the gap between ground-truth score function and the estimates: $L_{\text{score-matching}} = \mathbb{E}_{p(x)}[\|\nabla_x \log p(x) - s_\psi(x)\|\|_2^2]$. Since the objective is essentially regression with $l_2$ loss, the loss function does not require any assumptions on the parameterized function $s_\psi$, unlike the energy-based model LeCun et al. (2006) in which strong regularization is often required for stable training. These advantages make it possible to model a complex behavior policy $\beta$ with the high fidelity that offline RL requires. We briefly examine the ability of the score-based generative model using four discontinuous multi-modal distributions, and the results are shown in Figure A.1. In all four cases, the inferred probability distribution is very sharp, and its log probability resembles the penalty function we proposed.

## A.2 IMPLEMENTATION DETAILS

The implementation of ARQ consists of four steps: score-based generative model $s_\psi$ learning, sampling, Q-learning, and optional explicit policy $\pi_\phi$ training. For the score-based generative model, sampling, and likelihood computation, we generally follow the implementation of (Song et al., 2021) which trains a time-dependent neural network that approximates the reverse-time stochastic differential equations (SDE); a neural network learns to reverse the progressive diffusion process that turns a data point into random noise. Specifically, we use a value-preserving SDE (VPSDE) with a neural network having residual connections (He et al., 2015) with the embedding size of 256, and we stack 3 residual blocks. We use `swish` for nonlinearity. Since our target distribution is conditional (i.e., conditioned on a state) unlike the original formulation, we extend a time-dependent neural network to be a function of a state, an action, and time. We train the network with Adam with a learning rate of 1e-4 and batch size of 512, and we apply an exponential moving average with an average coefficient of 0.999. Since the number of training data samples varies across datasets, we tested the different number of training iterations and ensembles to prevent overfitting. For training iterations, we tested 150,000, 300,000, and 1 million steps, and for ensemble, we tried single and 3 ensemble models. For ensemble training, we train each model with different random samples from the same data pool. The dataset-specific hyperparameters are shown in Table A.1.

The second step of the ARQ algorithm is prepopulating samples for value learning. We use a predictor-corrector algorithm (Song et al., 2021) to generate samples by solving a reverse SDE. Again, we followed the implementation of Song et al. (2021), which uses the Euler-Maruyama method as a predictor and Langevin dynamics as a corrector. We discretize the time domain [1e-3,1] of



Figure A.1: The probability estimated by the trained score-based generative model for complex, discontinuous, and multi-modal conditional distributions $\beta(a|s)$. The state and action spaces are one-dimensional, and the conditional probability estimated using the model $s_\psi$ is represented with a color map. Red x marks represent the training samples.

Table A.1: Dataset-specific hyperparameters used in training a score function $s_\psi$.

| | | D4RL | | | Robomimic | |
| | Locomotion | Kitchen | Adroit | AntMaze | MG | PH, MH |
|---|---|---|---|---|---|---|
| Training Iterations | 1,000,000 | 300,000 | 300,000 | 1,000,000 | 1,000,000 | 150,000 |
| # Ensembles | 1 | 1 | 3 | 1 | 3 | 3 |

$s_\psi$ into 500 steps, and we execute a single corrector step for every predictor step. For Langevin dynamics, we dynamically adjust the noise scale using the norm of the score; we use a score-to-noise ratio of 0.16 as used in Song et al. (2021). Since the suggested penalty function is defined based on its probability $\beta(a|s)$, we need to compute the likelihood of generated samples using $s_\psi$. For the likelihood computation, we use an instantaneous change-of-variable formula on top of the ordinary differential equation induced from the SDE. To solve the inverse problem of ODE, we use the RK45 algorithm of scipy. For the detailed formulation and algorithmic detail regarding SDE, we refer to the original paper (Song & Ermon, 2019; 2020; Song et al., 2021) or our code. We use $\epsilon = e^{-5}$ across all experiments, and we drop any samples that show a lower probability than the given threshold $\epsilon$. We generate 30 samples for both ARQ and the implicit policy using $s_\psi$.

For ARQ training, we use an MLP with 2 layers and 256 activation nodes to parameterize the value function $Q_\theta$, and we apply `ReLU` nonlinear activation. Also, we shape the reward function of datasets following (Kostrikov et al., 2021b); for the locomotion tasks, the reward is normalized by multiplying the ratio of returns between the worst and the best trajectories in the dataset, and for the antmaze tasks, the reward is set to -1 except the goal state. Similarly, we densify the reward function of Robomimic using the same technique used for the antmaze. For stability, we train two Q functions and a slowly moving target network with a polyak coefficient of 0.995. We train 1 million timesteps using Adam optimizer and batch size of 512. We perform a rough random search with the following range of values for the following hyperparameters: $K \in [3, 6, 9]$, a learning rate $\in$ [3e-4,1e-4]. The chosen hyperparameters are shown in Table A.2. For the $Q^\beta$ used in the ablation study, we use the same parameterization of the value function, but we train a single Q function with a slowly moving target network. The value function is trained 1 million time steps using Adam with a learning rate of 1e-4 and batch size of 512.

Table A.2: Dataset-specific hyperparameters used or in ARQ.

| | | D4RL | | | Robomimic | |
| | Locomotion | Kitchen | Adroit | AntMaze | MG | PH, MH |
|---|---|---|---|---|---|---|
| Learning rate | 3e-4 | 1e-4 | 1e-4 | 3e-4 | 1e-4 | 1e-4 |
| $K$-th | 9 | 9 | 9 | 3 | 9 | 9 |
| Reward | Normalized | Original | Original | -1 except goal | -1 except goal | |

For the implicit policy that is based on the samples of $s_\psi$, we first generate 30 samples using $s_\psi$, then we resample an action from the categorical distribution that treats advantages as logits. Similarly, for the explicit policy, we train a policy using weighted behavior cloning (Peng et al., 2019) where the weight is computed using the advantage. We use state-independent stochastic policy used in (Kostrikov et al., 2021b) for the locomotion, kitchen, and adroit tasks of D4RL datasets, which predicts the mean $\mu(a|s)$ and the state-independent standard deviation $\sigma(a)$ of a Gaussian distribution. We use a 2-layer MLP having 256 hidden units with `ReLU` activations. For the antmaze tasks, we use the deterministic policy that omits the standard deviation prediction. Similarly, a deterministic policy is used for the Robomimic datasets, but we use dense Resnet blocks for the parameterization. We stack 4 ResNet blocks, each of which has 2048 embedding dimensions. We tried Gaussian Mixture Network (GMM) as suggested in (Mandlekar et al., 2021), but we could not replicate the reported performance in the BC setting. For D4RL and Robomimic, we train a policy for 1 million and 300,000 steps respectively, using the Adam optimizer with a learning rate of 3e-4. The key hyperparameter for the policy is the temperature term $\alpha$. We tested the following range of values: [0.1, 1.0, 10.0, 30.0], and we display the chosen values in Table A.3 along with other hyperparameters.

Table A.3: Dataset-specific hyperparameters used in the implicit policy and the explicit policy $\pi_\phi$.

| | D4RL | | | | Robomimic | |
| | Locomotion | Kitchen | Adroit | AntMaze | MG | PH, MH |
|---|---|---|---|---|---|---|
| $\alpha$ for $Q^\beta + s_\psi$ | 1 | 1 | 10 | 10 | 1 | 0.1 |
| $\alpha$ for ARQ $+ s_\psi$ | 1 | 1 | 10 | 10 | 1 | 0.1 |
| $\pi_\phi$ | State-independent stochastic | | | Det. | Det.-ResNet | |
| Training Iterations | 1,000,000 | | | | 300,000 | |
| $\alpha$ for ARQ $+ \pi_\phi$ | 1 | 1 | 10 | 10 | 10 | 0.1 |

## A.3 Full Experiment Results

We display the full experiment results in Table A.4.

Table A.4: Performance of prior methods and ours on D4RL(Fu et al., 2020) and RobomimicMandlekar et al. (2021) datasets. Each number represents the performance relative to a random policy as 0 and an expert policy as 100. Unless noted as (ours) or (repro.), all the numbers are borrowed from Kostrikov et al. (2021b), Fujimoto & Gu (2021), and Mandlekar et al. (2021). The numbers generated by us are averaged over 3 different random seeds. The standard deviations of multiple runs are also displayed.

| | | Without reward | | | | With $Q^\beta$ | | | | | With reward / value function | | | | | | |
|---|---|---|---|---|---|---|---|---|---|---|---|---|---|---|---|---|---|
| | | BC | Impl. BC | $s_\psi$ (ours) | $\sigma$ | One-step | $Q(\beta)$ $+s_\psi$ (ours) | $\sigma$ | DT | AWAC | TD3 +BC | CQL | IQL | ARQ $+\pi_\phi$ (ours) | $\sigma$ | ARQ $+s_\psi$ (ours) | $\sigma$ |
| expert-v2 | hopper | 112 | 110 | 89 | (5.1) | | 99 | (2.1) | | 85 | 108 | 111 | 110 | 111 | (0.2) | 98 | (0.8) |
| | walker | 56 | 82 | 106 | (0.6) | | 107 | (0.4) | | 57 | 110 | 104 | 110 | 109 | (0.4) | 108 | (0.1) |
| | halfchtah | 105 | 78 | 81 | (0.2) | | 84 | (1.1) | | 79 | 97 | 82 | 95 | 94 | (0.2) | 85 | (0.5) |
| medium-v2 | hopper | 53 | 75 | 38 | (1.3) | 60 | 55 | (0.6) | 68 | 57 | 59 | 59 | 66 | 61 | (0.4) | 58 | (0.8) |
| | walker | 75 | 15 | 63 | (1.6) | 82 | 79 | (0.6) | 74 | 72 | 84 | 73 | 78 | 81 | (0.7) | 81 | (0.2) |
| | halfchtah | 43 | 35 | 40 | (0.3) | 48 | 43 | (0.0) | 43 | 44 | 48 | 44 | 47 | 45 | (0.3) | 45 | (0.2) |
| med.-exp.-v2 | hopper | 53 | 28 | 45 | (0.4) | 103 | 88 | (4.1) | 108 | 56 | 98 | 105 | 92 | 110 | (0.9) | 93 | (2.5) |
| | walker | 108 | 16 | 74 | (2.8) | 113 | 107 | (0.2) | 108 | 75 | 110 | 109 | 110 | 109 | (0.5) | 107 | (0.4) |
| | halfchtah | 55 | 35 | 46 | (0.4) | 93 | 77 | (1.8) | 87 | 43 | 91 | 92 | 87 | 91 | (0.7) | 82 | (0.3) |
| med.-rep.-v2 | hopper | 18 | 14 | 11 | (0.3) | 98 | 62 | (0.9) | 83 | 37 | 61 | 95 | 95 | 81 | (24.2) | 78 | (3.2) |
| | walker | 26 | 10 | 20 | (0.3) | 50 | 69 | (2.0) | 67 | 27 | 82 | 77 | 74 | 66 | (7.0) | 71 | (1.2) |
| | halfchtah | 37 | 25 | 26 | (0.8) | 38 | 41 | (0.3) | 37 | 41 | 45 | 46 | 44 | 42 | (0.3) | 41 | (0.1) |
| human-v0 | pen | 64 | 84 | 76 | (6.0) | | 73 | (1.9) | | | | 38 | 72 | 44 | (5.2) | 74 | (1.3) |
| | door | 2 | 14 | 9 | (0.9) | | 11 | (1.4) | | | | 10 | 4 | 0 | (0.4) | 10 | (4.9) |
| | relocate | 0 | 0 | 0 | (0.0) | | 0 | (0.0) | | | | 0 | 0 | 0 | (0.0) | 0 | (0.1) |
| | hammer | 1 | 1 | 2 | (0.4) | | 5 | (1.2) | | | | 4 | 1 | 1 | (0.7) | 6 | (1.0) |
| cloned-v0 | pen | 37 | | 26 | (4.3) | 60 | 57 | (2.3) | | | | 39 | 37 | 48 | (7.1) | 55 | (2.7) |
| | door | 0 | | 2 | (0.4) | 0 | 11 | (2.0) | | | | 0 | 2 | 1 | (0.4) | 12 | (4.4) |
| | relocate | 0 | | 0 | (0.0) | 0 | 0 | (0.0) | | | | 0 | 0 | 0 | (0.0) | 0 | (0.0) |
| | hammer | 1 | | 1 | (0.2) | 2 | 3 | (0.3) | | | | 2 | 2 | 1 | (0.3) | 4 | (2.6) |
| kitchen-v0 | complete | 65 | 85 | 74 | (3.7) | | 75 | (1.2) | | | | 44 | 63 | 37 | (14.2) | 77 | (1.8) |
| | partial | 38 | 38 | 45 | (2.7) | | 59 | (4.9) | | | | 50 | 46 | 50 | (5.0) | 70 | (2.7) |
| | mixed | 52 | 38 | 51 | (1.1) | | 52 | (1.1) | | | | 51 | 51 | 39 | (9.4) | 57 | (2.9) |
| antmaze-v0 | umaze | 55 | | 58 | (2.1) | 64 | 81 | (4.5) | 59 | 57 | 79 | 74 | 88 | 97 | (0.8) | 94 | (1.7) |
| | umaze-div. | 46 | | 61 | (1.4) | 61 | 62 | (3.3) | 53 | 49 | 71 | 84 | 62 | 62 | (12.1) | 58 | (7.0) |
| | med.-play | 0 | | 1 | (0.5) | 0 | 25 | (13.3) | 0 | 0 | 11 | 61 | 71 | 80 | (8.3) | 69 | (6.6) |
| | med.-div. | 0 | | 1 | (0.5) | 0 | 45 | (5.3) | 0 | 1 | 3 | 54 | 70 | 82 | (6.1) | 65 | (15.6) |
| | large-play | 0 | | 0 | (0.0) | 0 | 1 | (0.5) | 0 | 0 | 0 | 16 | 40 | 37 | (17.7) | 18 | (1.7) |
| | large-div. | 0 | | 0 | (0.0) | 0 | 1 | (0.5) | 0 | 1 | 0 | 15 | 48 | 58 | (6.2) | 23 | (5.0) |

| | | BC* | | $s_\psi$ (ours) | | $Q(\beta)$ $+s_\psi$ (ours) | | | | BCQ* | CQL* | IQL (repro.) | ARQ $+\pi_\phi$ (ours) | | ARQ $+s_\psi$ (ours) | |
|---|---|---|---|---|---|---|---|---|---|---|---|---|---|---|---|---|
| machine gen. | lift | 65 | | 29 | (2.4) | 86 | (0.5) | | | 91 | 64 | 79 | 79 | (1.2) | 82 | (0.5) |
| | can | 65 | | 19 | (2.4) | 55 | (6.5) | | | 75 | 1 | 62 | 76 | (0.5) | 60 | (1.2) |
| pro. human | lift | 100 | | 99 | (0.8) | 100 | (0.5) | | | 100 | 93 | 58 | 100 | (0.0) | 98 | (0.0) |
| | can | 95 | | 95 | (0.8) | 93 | (2.4) | | | 89 | 38 | 26 | 92 | (2.2) | 95 | (0.8) |
| | square | 79 | | 66 | (2.9) | 72 | (4.0) | | | 50 | 5 | 24 | 44 | (2.2) | 69 | (3.4) |
| | transport | 17 | | 27 | (2.5) | 28 | (8.7) | | | 7 | 0 | 1 | 29 | (5.0) | 30 | (5.4) |
| | tool-hang | 29 | | 70 | (4.2) | 64 | (2.9) | | | 0 | 0 | 3 | 3 | (1.7) | 71 | (5.9) |
| multi. human | lift | 100 | | 96 | (1.7) | 94 | (3.6) | | | 100 | 57 | 51 | 99 | (1.4) | 95 | (2.5) |
| | can | 86 | | 84 | (1.2) | 89 | (1.6) | | | 63 | 22 | 25 | 90 | (1.2) | 86 | (1.7) |
| | square | 53 | | 44 | (1.7) | 51 | (4.9) | | | 14 | 1 | 12 | 31 | (4.2) | 51 | (4.8) |
| | transport | 11 | | 15 | (3.1) | 17 | (2.6) | | | 3 | 0 | 0 | 16 | (4.9) | 13 | (1.7) |

⋆ represents that the best performance during training iterations is picked a posteriori.

