# OpenReview forum: "Know Your Boundaries: The Advantage of Explicit Behavior Cloning in Offline RL"
_ICLR.cc/2023/Conference — Submitted to ICLR 2023_

### Official Review · Reviewer_8gEu · 2022-10-18

**Confidence:** 3
**Correctness:** 2
**Technical Novelty And Significance:** 3
**Empirical Novelty And Significance:** 3
**Recommendation:** 3

**Clarity, Quality, Novelty And Reproducibility:**

**Novelty** -- I'll decompose the method into (1) a new offline RL method and (2) using a score-based generative model for estimating the behavioral policy.
1. The precise form of the offline RL algorithm seems novel, but it is very similar to prior work; roughly speaking, it's a combination of BCQ and AWR. I think these similarities should be awknowledged and studied. If they are important, it'd be great to demonstrate that (e.g., compare AWR, BCQ, and the proposed methods, all using exactly the same hyperparameters and without using any score-based generative models). If they aren't important, I'd recommend ditching this contribution and just using one of the prior methods (no need to create another method if the existing methods work fine).
2. Using a score-based generative model is relatively novel. One recent paper [1] is fairly similar, and should be mentioned; but, given that it was released in August, it seems fine to argue that that was concurrent work.

**Quality** -- There are a few issues with both the writing and the experiments (see details above).

**Reproducibility** -- The Appendix seems to contain the relevant experimental details, and the supplemental material includes code.


[1] https://arxiv.org/abs/2208.06193

**Strength And Weaknesses:**

Strengths
* Empirical results -- The empirical results are quite strong. If it is the case that these benefits are coming solely from the more powerful generative model (rather than the new offline RL algorithm), I think that would be a pretty interesting result. One reason it's surprising is that the policies that collect these datasets are usually pretty simple, so it's counterintuitive that we'd need really powerful density models to learn the behavioral policy.
* It's great that the paper released code!

Weaknesses
* Relationship with prior work -- The paper does not sufficiently discuss the similarities with prior methods. As noted below, I think many of the claims about prior work are not true, and the proposed method is very similar to prior work. The strong empirical results suggest that this is simply a matter of writing -- situating the method relative to prior works, noting that the method is actually very similar to prior work (perhaps don't claim that as a contribution), and arguing that the score-based generative model is the important contribution. One way this could be done is to *not* propose any new method, but show that using a more powerful model for estimating $\beta(a \mid s)$ can be used to improve *existing* offline RL methods. In fact, the $s_\psi$ and $Q(\beta) + s_\psi$ columns in Table 2 already do this.
* Correctness -- I'm not sure about some of the arguments in the paper (see details below). Again, I think this is purely a problem with writing, but I suspect that many parts of the paper will have to be rewritten.
* The proposed method seems to require per-domain hyperparameter tuning.


There are a number of claims that I'm unsure about.
* "behavioral cloning cannot extract a good policy" -- This seems to contradict the earlier claim in the introduction that certain off-policy methods "fall short compared to simple behavioral cloning."
* "high-fidelity behavioral cloning has not been achieved" -- Cite. Note that IBC and [1] might be counterexamples.
* "despite the need in offline RL for precist estimation of behavior policy (Levine)" -- Where does (Levine,) argue this?
* "have only been utilized with heuristics or proxy formulations" -- I don't think this is true. Many prior works (including those cited) do provide theoretical justification for the proposed methods.
* "which was previously considered unnecessary or infeasible (...) " -- I don't think these citations say this. Also, this countradicts the claim in the previous paragraph that (Levine 2020) argues for high-fidelity behavioral cloning.
* Related work -- I think some of the categorization here isn't accurate. For example, some of the methods in the first paragraph do "value learning," and arguably all of the methods in the first paragraph "try to resolve the distribution shift problem"
* Related work, method -- I'd recommend acknowledging the large body of offline RL work that came before 2019. Much of it is referred to as "batch RL" rather than "offline RL".
* "$p(s, a) = Q_\theta - Q^{\pi_\phi}$" -- This is a neat idea, but seems different from what the rest of the paragraph discusses. Prior methods that estimate uncertainty are estimating the _magnitude_ of this difference, but not the _sign_ of the difference.
* "it has not been thoroughly investigated" -- I don't think this is true. There is a fair bit of work in this area. For example, BRAC compares a bunch of different divergence measures.
* "can only be understood as ad-hoc methods" -- I don't think this is true. Many offline RL papers (including those cited) include theoretical motivations for the proposed method.
* "where $\pi_p(a \mid s) = \text{softmax}(-p(s, a))$ -- Is the $p(s, a)$ here different from the one defined in paragraph 3 of the method section?
* Theorem 1 -- I think this is trivial, as KL control is a special case of regularized policy iteration (see, e.g., [2]).
* "K-th is an operator" -- Is this only well defined for discrete actions?
* Eq. 6 -- This doesn't integrate to one. I think the numerator is missing a term $\delta(a \in \text{supp}(\beta))$.

Minor writing comments
* Abstract -- it's a bit unclear how the proposed method is different from prior methods that add explicit behavioral cloning (e.g., BCQ, TD3+BC).
* "warrants consideration" -- the passive voice is to be avoided.
* "attempts to use an explicitly trained behavioral cloning model in offline RL" -- I think there are many more citations for this.
* "Offline RL is about exploitation" -- I'm not sure this is true.
* "are preferred over" -- Who prefers them? Add a citation
* "utilizes the half side of the information" -- I didn't understand this.
* "support the idea" -- What idea?
* "stabilizing the learning" -- Does this refer to the learning of value-based methods?
* "value-based methods, which can be categorized as" -- I think "which" is referring to the wrong noun
* "instantiating it can" -- What does "it" refer to?
* "straightforwardly" -- Cut.
* $s_\psi(s, a)$ -- It'd be good to formally state what this is defined to be (the gradient of ...)
* AWR -- Cite Neumann 2008 [3], which proposed a similar method much earlier.
* Table 2 -- I'd recommend positioning this Table one page earlier. Clarify what $s_\phi$ is.
* Figure A.1 -- It's hard to see the predicted density because it's occluded by the red Xs

[1] https://arxiv.org/abs/2208.06193

[2] https://proceedings.mlr.press/v97/geist19a/geist19a.pdf

[3] https://proceedings.neurips.cc/paper/2008/file/f79921bbae40a577928b76d2fc3edc2a-Paper.pdf


**Summary Of The Paper:**

This paper proposes an offline RL method that combines a BCQ-style critic update (sample $k$ actions from $\beta(a \mid s)$ and take the max) and an AWR-style actor update. One important detail is that the behavioral policy $\beta(a \mid s)$ is fit using a much more powerful model class (a score-based generative model) than prior work. Empirically, the proposed method outperforms IQL and other baselines on the majority of tasks from D4RL and robomimic.

**Summary Of The Review:**

Overall, I think this paper is studying an interesting problem (how powerful explicit behavioral models can be used for offline RL). While the current experiments and writing have a number of limitations (see above),  I think that a revised version of the paper would make for a strong submission to a future conference.

---

> ### Author Response · Authors · 2022-11-14
> **To reviewer 8gEu #3**
>
> > Theorem 1 -- I think this is trivial, as KL control is a special case of regularized policy iteration (see, e.g., [2]).
>
> Thank you for the pointer, and we will add more references to the previous works that have shown the connection between regularized MDP and KL control.
>
> Yet, we would like to emphasize that the goal of the suggested theorem is to examine other offline RL algorithms under the same view of KL-regularization. It becomes possible because of the broad expressivity of the proposed DQP framework, which allows defining offline RL algorithms in terms of a penalty function. Therefore, we argue that the contribution of this work more heavily resides in Section 4.2 and Table 1, rather than the theorem itself.
>
> > Eq. 6 -- This doesn't integrate to one. I think the numerator is missing a term δ(a∈supp(β))
> > "K-th is an operator" -- Is this only well defined for discrete actions?
>
> Thank you for pointing out the notational problems. ARQ approximates the support set of the behavior policy $\beta$ with finite samples from the trained model $s_\psi$. Because of this, we can use the $K$-th operator even for the continuous case, and Eq. 6 can integrate to 1. To resolve the issue, we change the notation $supp(\beta)$ to $\bar{supp}(\beta)$ and make it clear that we are using a discrete set consisting of finite samples from $\beta$.
>
> > It's a bit unclear how the proposed method is different from prior methods that add explicit behavioral cloning (e.g., BCQ, TD3+BC).
>
> We rewrite the Related Works section to make the difference between ours and prior methods that explicitly model a behavior policy, such as BCQ or BRAC. Please checkout the updated manuscript, and let us know if the paper can get enhanced more.
>
> One important thing to mention is that in our categorization, [TD3+BC] (and the concurrent work [1], which are largely based on [TD3+BC] is not a method that explicitly clones a behavior policy. The prior work did include the behavior cloning loss in their policy training, the loss term is linearly combined with ordinary actor-loss of TD3, and therefore, behavior policy $\beta$ itself is never modeled.
>
> Interestingly, our theoretical framework, DQP, can envelop their approach. The penalty function of TD3+BC is: $-\alpha (a - \beta(s))^2$. This form of penalty function shares the similar problem to BRAC-KL, which prefers a more frequently executed action in the dataset; intuitively, when $\alpha$  is set to very large numbers, TD3+BC will almost behave like behavior cloning. We updated Table 1 including  the penalty function of [TD3+BC], and added the discussion in Section 4.2.
>
> > The proposed method seems to require per-domain hyperparameter tuning.
>
> While there are many tunable hyperparameters, the training of the generative model is a fully supervised setup in which hyperparameters are easily tunable without accessing an MDP. In our case, we just used domain-agnostic methods (e.g. simple train-validation splitting) to check the overfitting of generative models, and we adjusted the number of training iterations and ensembles accordingly. Also, the log probability threshold can be set by computing the log-likelihood on the validation set. We can actually observe some performance boost by separately tuning the threshold, but we omit the tuning in our final evaluation to show the robustness of our method. The hyperparameter K can also be tuned using training statistics, similar to [CQL-training-workflow]. That is, we can use smaller K as long as the training is stable. Similarly, we observed higher performance using different K for each dataset in our early experiments, but we did not include such results in the paper.
>
> In our experience, the only hyperparameter that needs careful tuning is $\alpha$. This is understandable because it depends on the scale of the original reward function, and a similar hyperparameter is also shown across different offline RL methods. Therefore, we do not see this as a weakness specific to our method.
>
> * [CQL-training-workflow]: Aviral Kumar et al., A workflow for offline model-free robotic reinforcement learning, CoRL 2021.
>
> > One recent paper [1] is fairly similar, and should be mentioned; but, given that it was released in August, it seems fine to argue that that was concurrent work.
>
> Thank you for the pointer to the concurrent work. We updated the manuscript and cited the concurrent work that utilizes a strong generative model similar to ours. That being said, our work should be evaluated independently of concurrent works that were not yet published at the time of our submission.
>
> Reviewer 8jQf also raised a similar question asking the relationship between ours and [1], and we provided a detailed comparison including potential weakness of the other work in the response. Please find our answer in the response for Reviewer 8jQf.

---

> ### Author Response · Authors · 2022-11-14
> **To reviewer 8gEu #2**
>
> > I think some of the categorization here isn't accurate. For example, some of the methods in the first paragraph do "value learning," and arguably all of the methods in the first paragraph "try to resolve the distribution shift problem"
>
> > I'd recommend acknowledging the large body of offline RL work that came before 2019. Much of it is referred to as "batch RL" rather than "offline RL".
>
> We reorganized the related works section to reflect the reviewer’s concerns; we added several previous works and recategorized them. Also, we tried to posit our work in the context of motivating previous works like BCQ while making the distinction clearer. Please let us know if the new categorization still has a problem, or if we miss other important previous works.
>
> > "p(s,a)=Qθ−Qπϕ" -- This is a neat idea, but seems different from what the rest of the paragraph discusses. Prior methods that estimate uncertainty are estimating the magnitude of this difference, but not the sign of the difference.
>
> The introduced penalty function is more like a general statement that expresses the ultimate goal of offline RL under the DQP framework. This is because the given penalty function is an oracle defined on top of the ground truth $Q^{\pi_\phi}$, which is the goal of policy evaluation. The oracle-ness of the given penalty function implies that the penalty function is infeasible, and therefore, we do not suggest an idea or a novel method by introducing that penalty function.
>
> The core message of the paragraph is to provide a high-level, abstractive view on (1) what offline RL methods should do and (2) what current algorithms are doing; what should do is trying to approximate the given penalty function (in both magnitude and sign per se), and what most offline RL methods do is trying to approximate the magnitude only and use the magnitude in a pessimistic way (assume the sign is always plus == assume predictions are overestimated). This is because, without environment-specific assumptions, both magnitude and sign cannot be estimated accurately.
>
> We hope this explanation clarifies the question. Please let us know if you have further questions or specific suggestions to enhance the manuscript. We will be happy to answer questions and incorporate the suggestions!
>
> > "it has not been thoroughly investigated" -- I don't think this is true. There is a fair bit of work in this area. For example, BRAC compares a bunch of different divergence measures.
>
> We acknowledge the reviewer’s concern that the statement may be too strong. We toned down the sentence incorporating the concern. The change is the following: “DQP provides a unified way to represent different offline RL algorithms in terms of penalty functions. However, in order for a unified perspective to provide a better understanding of offline RL algorithms and thereby be useful for developing better offline RL algorithms, it is necessary to investigate the effect of the penalty function under the policy iteration scheme.”
>
> That being said, the difference between the prior works and ours is noteworthy. BRAC indeed compares several different divergence measures, but the analysis could be limited in that BRAC only considers penalty functions that can be representable in a divergence between two policies, while the suggested DQP framework can incorporate a general form of penalty function that does not involve divergence between two policies, such as Anti-Exploration, PLOFF, CQL, etc. as shown in Table 1.
>
> > "where πp(a∣s)=softmax(−p(s,a)) -- Is the p(s,a) here different from the one defined in paragraph 3 of the method section?
>
> $\pi_p$ is a policy induced by a penalty function; that is, the equation represents the definition of $\pi_p$. To avoid confusion, we change the notation to $:=$.
>
> Please note that each different penalty function will have a different form of $\pi_p$, which can provide some intuition about how each penalty function regularizes the learning under the policy iteration scheme. The discussion related to different $\pi_p$ are given in Section 4.2 along with the penalty function of ours  in Equation (4).

---

> ### Author Response · Authors · 2022-11-14
> **To reviewer 8gEu #1**
>
> We sincerely appreciate the reviewer’s scrutiny to make the paper more precise and correct. Since most concerns are related to the writing, we rewrote many parts of the paper incorporating all the provided comments. We corrected the phrases that might cause over-interpretation, and added paragraphs that make our contribution more distinct compared to prior works. Especially, the Related Works section is rewritten. We also provided a detailed explanation in the response to clarify some confusion. Please let us know if there are parts that need extra revision. We will be happy to revise more. If the update addresses the reviewer’s concern, we kindly ask the reviewer to reassess the paper in light of the update.
>
> > "behavioral cloning cannot extract a good policy" -- This seems to contradict the earlier claim in the introduction that certain off-policy methods "fall short compared to simple behavioral cloning."
>
> We apologize for the confusion. The core message we wanted to deliver from the paragraph is that BC and offline RL methods can be *complementary*; BC would obviously fail to learn a good policy if the given dataset contains only suboptimal trajectories, but the learning is often more stable than offline RL methods, and thereby it sometimes shows a good empirical performance. In contrast, offline RL methods could extract a good policy even from a set of suboptimal trajectories theoretically, but it is often difficult to tune the algorithms, and this often results in lower performance than BC baselines.
>
> The last sentence is added to emphasize the disadvantage of a simple BC approach, which cannot surpass the performance of the provided dataset. However, we acknowledge that the sentence can be confusing. We believe the message is much clearer without the last sentence, so we eliminate it in the revision. Please let us know if the paragraph needs extra revision.
>
> > "high-fidelity behavioral cloning has not been achieved" -- Cite. Note that IBC and [1] might be counterexamples.
>
> To make the sentence clear, we change the sentence as follows: “high-fidelity generative models have not been integrated with offline RL algorithms.”
>
> Note the difference between ours and IBC: IBC is solely a behavior-cloning algorithm that only partially uses reward signals without training a value function while our method integrates BC into a value function. To provide some context, we actually started this research to extend IBC and make it more offline RL-ish. Yet, we couldn’t make progress with the energy-based model due to its training instability, so we decided to test out better generative models, and we ended up with the current algorithm. We hope this extra bit helps you to situate our method related to others.
>
> > "despite the need in offline RL for precise estimation of behavior policy (Levine)" -- Where does (Levine,) argue this?
>
> > "which was previously considered unnecessary or infeasible (...) " -- I don't think these citations say this. Also, this contradicts the claim in the previous paragraph that (Levine 2020) argues for high-fidelity behavioral cloning.
>
> The part that we referred is the following:
>
> “This means that the performance of these algorithms is limited by the accuracy of estimation of the behavior policy, which may be hard in scenarios with highly multimodal behaviors, as is the case in practice with real-world problems. (...) Methods that enforce the constraint implicitly, using only samples and without explicit behavior policy estimation, are a promising way to alleviate this limitation.” (Page 25, second paragraph)
>
> Yet, we admit that the way we used the original text might be misleading, or extrapolating the idea more than that is originally intended by the author. We changed the phrasing regarding the reviewer’s concern.
>
> > "have only been utilized with heuristics or proxy formulations" -- I don't think this is true. Many prior works (including those cited) do provide theoretical justification for the proposed methods.
>
> > "can only be understood as ad-hoc methods" -- I don't think this is true. Many offline RL papers (including those cited) include theoretical motivations for the proposed method.
>
> We agree that many prior works have a solid theoretical ground as the reviewer stated, and we did not intend to say that prior methods are not theoretically justifiable. Yet, our augment is more focused on the “implementation” side of the algorithm. For instance, BEAR also implements the same support-set constraint and they provide a beautiful theory in their paper. Yet, they resort to MMD distance for their implementation as a ***proxy*** to instantiate the support-set constraint. In contrast, our method tries to directly instantiate the support set constraint by explicitly modeling the behavior policy, and it empirically provides better results.
>
> To avoid further confusion, we changed the phrase, and it can be checked through the uploaded manuscript.

---

> > ### Comment · Reviewer_8gEu · 2022-11-14
> > **Reviewer response**
> >
> > Dear authors,
> >
> > Thank you for the detailed response. The revisions to the paper have made the paper stronger. Reading through responses 1/2/3, I'm not sure they address my main concern: that it's unclear whether this paper is about a new offline RL algorithm or a new policy architecture for offline RL. A secondary concern is that I'm still unsure about some of the revisions (I've only double checked the revisions in the abstract/intro). Taken together, I am still reticent to accept the paper.
> >
> > Comments about the revisions to the introduction:
> >
> > > Where does (Levine,) argue this?
> >
> > Thanks for the reference to the precise paragraph. However, (Levine,) doesn't seem to provide any evidence (citations or experiments) for this claim. It seems likely that the claim is true, but I just haven't seen the evidence yet.
> >
> > > Florence et al. (2021) have tried an energy-based generative model, but the proposed method is an imitation learning that does not incorporate a value function
> >
> > Conditional behavioral cloning methods, like Florence 2021, do _implicitly_ use a value function [1].
> >
> > > Second, the trained behavior cloning models have only been utilized with certain limited forms,
> >
> > I'm not sure that this claim is correct. There has been a large amount of work (including in Wu et al, 2019!) that uses a variety of different ways of using the learned behavioral policy.
> >
> > [1] https://arxiv.org/abs/2206.03378

---

> > > ### Author Response · Authors · 2022-11-16
> > > **Second response to reviewer 8gEu #3**
> > >
> > > > Conditional behavioral cloning methods, like Florence 2021, do implicitly use a value function [1].
> > >
> > > We would like to encourage the reviewer to review our Related Work, specifically, the recategorization of offline RL methods. To summarize here briefly, our categorization of offline RL is three-folded: (1) the offline RL methods that are done in two steps; first behavior cloning $\beta$ explicitly, and value learning + policy learning with regularization using trained $\beta$. (2) the offline RL methods that are done in a single step; do value learning + policy learning w/ some implicit regularization to keep a policy not too far from $\beta$. (3) BC method that does not incorporate value learning. We compared our method to (Florence et al., 2021) to emphasize the importance of the ***value learning*** step since what the methods belonging to the third category do is the first step of the methods of the first category, and the extra step would make a huge performance boost as we shown in our experiments (comparison between $s_\psi$ and $ARQ+s_\psi$).
> > >
> > > We acknowledge that our categorization could be incomprehensive. However, we believe the comment of the reviewer is inaccurate since we clearly limited our discussion specifically to Florence et al. (2021) in that sentence, but the reviewer brought up concurrent work to refute the sentence. Furthermore, considering that the work the reviewer mentioned can be classified as “implicit” offline RL methods following our categorization, it seems like there still exists a misunderstanding. To emphasize, it is important to acknowledge that we simply clone $\beta$ without any goal, reward, or value conditioning. The methods that learn a $\beta$ with some modification, regularization, or filtration belong to “implicit’ methods.
> > >
> > > > I'm not sure that this claim is correct. There has been a large amount of work (including in Wu et al, 2019!) that uses a variety of different ways of using the learned behavioral policy.
> > >
> > > We believe the sentence should be understood as a whole, rather than taking half of the sentence out of context. (Wu et al., 2019) certainly has a limited form in that they only tried *divergence between cloned policy and the actor policy*: $D(\beta | \pi)$. In comparison, our DQP framework allows a more general form of regularization based on how the penalty function is designed; it is less obvious how we can represent the CQL in the framework of (Wu et al., 2019) while DQP can incorporate CQL.

---

> > > > ### Comment · Reviewer_8gEu · 2022-11-16
> > > > **Revisions to related work address my concerns there**
> > > >
> > > > Dear authors,
> > > >
> > > > Thanks for clarifying this. The revisions to the related work section address my concerns there -- I now agree that it is accurate.

---

> > > ### Author Response · Authors · 2022-11-16
> > > **Second response to reviewer 8gEu #2**
> > >
> > > > Thanks for the reference to the precise paragraph. However, (Levine,) doesn't seem to provide any evidence (citations or experiments) for this claim. It seems likely that the claim is true, but I just haven't seen the evidence yet.
> > >
> > > We believe the argument made by (Levine et al.) is some shared tacit knowledge among offline RL researchers; especially at the moment when the paper of (Levine et al.) came out, the experimental results had indicated that implicit methods, such as CQL, were preferable since the error caused by modeling the behavior policy would be larger than the benefit we can achieve through explicit modeling of the behavior policy. Yet, in our work, we showed that the error from cloning the behavior policy can actually be diminished using strong generative models, and the results are something that has not been reported in offline RL communities. Therefore, we firmly believe this is a novel contribution that can benefit the community.
> > >
> > > Being said, we ran an additional experiment to examine the validity of the tacit knowledge that we overlooked before. We investigated the importance of a high-fidelity behavior cloned model via an ablation study. We consider two ways of ablation: first, we replace the score-based generative model with a gaussian mixture model, and second, we lower the fidelity of the score-based model by adding a random gaussian noise during training the generative model. For GMM, we used a mixture of 5 Gaussian, and for the noise experiments, we tried 3 noise scales $c=[0.5,1.0,2.0]$ where the std of the noise distribution is the scale $c$ multiplied by the std of actions in datasets. We train Q-value functions using ARQ with the ablated generative models, and we test the performance of both the parametric policy and the generative-model-based policy. The results averaged over 3 random seeds are shown below.
> > >
> > > |  |  | medium-|replay-|v2  | adroit- | human-v0 | kitchen |  |  |
> > > |---|---|---|---|---|---|---|---|---|---|
> > > |  |  | hopper | walker | halfcheetah | pen | door | complete | partial | mixed |
> > > | GMM | s | 24 | 28 | 35 | 68 | 6 | 78 | 38 | 47 |
> > > |  | ARQ + pi | 62 | 57 | 42 | 58 | 0 | 36 | 55 | 29 |
> > > |  | ARQ + s | 59 | 78 | 41 | 63 | 7 | 27 | 10 | 24 |
> > > | Original | s | 11 | 20 | 26 | 76 | 9 | 74 | 45 | 51 |
> > > |  | ARQ + pi | 81 | 66 | 42 | 44 | 0 | 37 | 50 | 39 |
> > > |  | ARQ + s | 78 | 71 | 41 | 74 | 10 | 77 | 70 | 57 |
> > > | Noise 0.5 | s | 6 | 5 | 4 | 37 | 1 | 6 | 14 | 13 |
> > > |  | ARQ + pi | 62 | 58 | 41 | 7 | 0 | 32 | 35 | 39 |
> > > |  | ARQ + s | 49 | 44 | 38 | 31 | 1 | 24 | 19 | 22 |
> > > | Noise 1.0 | s | 3 | 1 | 1 | 8 | 0 | 1 | 8 | 4 |
> > > |  | ARQ + pi | 63 | 0 | 2 | 1 | 0 | 0 | 0 | 0 |
> > > |  | ARQ + s | 49 | 1 | 1 | 10 | 0 | 1 | 7 | 5 |
> > > | Noise 2.0 | s | 1 | 0 | -1 | 0 | 0 | 2 | 6 | 3 |
> > > |  | ARQ + pi | 25 | 0 | 2 | 1 | 0 | 0 | 0 | 0 |
> > > |  | ARQ + s | 2 | 0 | -1 | 0 | 0 | 1 | 5 | 3 |
> > >
> > > We confirmed the argument that the accurate estimation of the behavior policy is critical for offline RL methods that explicitly perform BC; the performance degrades significantly in both ablations. The results are the evidence for the argument of (Levine et al.), and at the same time, it shows the novelty of our work that proves the potential of explicit behavior cloning in offline RL, which was previously reluctant. We hope this experiment helps the reviewer to clear out the doubt.

---

> > > > ### Comment · Reviewer_8gEu · 2022-11-16
> > > > **Quick clarification**
> > > >
> > > > Can the authors clarify what the rows are in the table above? (i.e., what is the objective function and model for "s", "ARQ + pi", and "ARQ + c"?) I Table 2 in the main text, I thought that $s_\psi$ was the score-based generative model -- does the first row of the table use a score-based generative model for the policy or a GMM policy?

---

> > > > > ### Author Response · Authors · 2022-11-18
> > > > > **Quick Clarification**
> > > > >
> > > > > We apologize for the confusion. The table could be confusing, especially the first row due to abuse of the notation; we extend the meaning of $s$ to refer to any parameterized network that can approximate $\beta(a|s)$. Therefore, the first group of rows (GMM group) represents the results when we explicitly model $\beta(a|s)$ using a gaussian mixture model, and the other following groups represent the results when we explicitly model $\beta(a|s)$ while we add a gaussian noise to action in a dataset. GMM is trained to maximize the log-likelihood.
> > > > >
> > > > > In each group of rows, the first row is simply testing the cloned policy (GMM or score-based model with noisy dataset), and the second and the third row shows the results of ARQ with the ablated behavior cloning models: Q-function is trained using Eq. 5 and the parameterized policy $\pi_\phi$ is trained via AWR on top of the Q-function (second row), or the policy is implicitly represented with trained the BC model $s$ and the Q-function (third row).

---

> > > > > > ### Comment · Reviewer_8gEu · 2022-11-18
> > > > > > **Continuing the discussion**
> > > > > >
> > > > > > Dear authors,
> > > > > >
> > > > > > Thanks for clarifying this! I've gone back and looked at the first 2 blocks of rows in the table. Are the following conclusions correct?
> > > > > >
> > > > > > 1. ARQ is better than behavioral cloning
> > > > > > 2. ARQ uses two policies: (1) one that mimics the dataset, and (2) another that tries to get higher returns than the dataset. Changing the architecture of (1) GMM --> score-based generative model often increases performance.
> > > > > > 3. Changing the architecture of (2) MLP(?) --> score-based generative model often increases performance.
> > > > > >
> > > > > > Also, are there any experiments comparing ARQ (without the score-based generative model) to prior offline RL methods (without the score-based generative model)?

---

> > > > > > > ### Author Response · Authors · 2022-11-19
> > > > > > > **Answers to the questions**
> > > > > > >
> > > > > > > > ARQ is better than behavioral cloning
> > > > > > >
> > > > > > > Yes. It is obvious since BC (whether it is score-based, energy-based or a simple MLP) does not exploit reward information.
> > > > > > >
> > > > > > > > ARQ uses two policies: (1) one that mimics the dataset, and (2) another that tries to get higher returns than the dataset.
> > > > > > >
> > > > > > > We assume the reviewer refers ARQ + $s$ as (1) and ARQ + $\pi$ as (2). Then, yes. One thing to note is that even in the case of (1) where we use the approximation of $\beta$, the resulting policy improves over $\beta$ since it is defined with the Q-function trained via ARQ.
> > > > > > >
> > > > > > > > Changing the architecture of (1) GMM --> score-based generative model often increases performance.
> > > > > > >
> > > > > > > Partially true; the change of architecture in (1) is made because we change the architecture for $\beta$. Again, we did not further ***train*** a new policy. We get a new policy by combining the approximated $\beta$ with the trained Q-function via ARQ (which also uses the approximation of $\beta$.)
> > > > > > >
> > > > > > > Given this information, Yes, the performance increase when we change the architecture for $\beta$ from GMM to a score-based generative model. Basically, the results indicate that accurate approximation of $\beta$ is beneficial.
> > > > > > >
> > > > > > > > Changing the architecture of (2) MLP(?) --> score-based generative model often increases performance.
> > > > > > >
> > > > > > > We believe that the reviewer focused on the comparison between ARQ + $\pi$ and ARQ + $s$. It is inaccurate to describe the comparison between the two as an architectural change. It is to check whether we need an extra step to train (or in some sense, extract) a parameterized policy from the Q-function trained via ARQ. The results indicate that it is often unnecessary to train a parameterized policy since well-approximation of $\beta$ can perform well.
> > > > > > >
> > > > > > > > Also, are there any experiments comparing ARQ (without the score-based generative model) to prior offline RL methods (without the score-based generative model)?
> > > > > > >
> > > > > > > GMM results (the first group of rows) can be interpreted as ARQ without score-based generative models, and the results can be directly comparable to other offline RL methods shown in Table 2 (or Table A.4).

---

> > > > > > > > ### Comment · Reviewer_8gEu · 2022-11-21
> > > > > > > > **Reviewer response**
> > > > > > > >
> > > > > > > > Thanks for these answers. I will take them into account during the final review.

---

> > > ### Author Response · Authors · 2022-11-16
> > > **Second response to reviewer 8gEu #1**
> > >
> > > Thank you for the prompt response. It seems like there are still doubts and concerns regarding the paper, and we hope this second response would address the issues. Please let us know if there are any remaining concerns or doubts.
> > >
> > > > It's unclear whether this paper is about a new offline RL algorithm or a new policy architecture for offline RL.
> > >
> > > The foremost message of our paper is to learn a behavior policy $\beta$ with strong generative models to constrain value learning. In fact, the proposed algorithm ARQ does not train a policy with a strong generative model while the estimated $\beta$ is mainly used to regularize the value learning; with the trained value function, we either train a feed-forward neural network as other implicit method does (ARQ + $\pi_\phi$) or use the estimated $\beta$ without explicitly training a parameterized policy (ARQ + $s_\psi$). Therefore, it is inaccurate to describe our contribution as suggesting a new policy architecture for offline RL.
> > >
> > > Certainly, the powerful model is crucial for strong empirical performance, especially when a task involves complex dynamics such as contact, or a dataset contains a few human demonstrations, which requires a policy to mimic a narrow, multimodal distribution defined with a few distinct demonstrations. However, when we compare the results between $Q(\beta) + s_\psi$ and $ARQ + s_\psi$, it is obvious that a regularized value function is also essential for strong performance. Considering that we also provide a justification for our use of trained $\beta$, we believe our argument on using strong generative models for the regularization of value function is well-supported, and the contribution is clear.
> > >
> > > As far as we are concerned, there hasn’t been a work to (1) use a strong generative model to (2) explicitly clone the behavior policy and then (3) use it to fulfill support-set constraints; BCQ and BRAC-KL or BRAC-Wasserstein explicitly clone the behavior policy, but they used VAE and did not sincerely instantiate the support-set constraint. BEAR also contains a similar problem due to the use of MMD-distance without explicitly cloning the behavior policy. TD3+BC has a loss term related to behavior cloning, but the meaning of the loss term is not interpretable as our instantiation of the support-set constraint. Even the concurrent work that uses diffusion models [1], our work is different from theirs because they are based on TD3+BC, which inherits the same problem of interpretation.
> > >
> > > > I'm still unsure about some of the revisions (I've only double checked the revisions in the abstract/intro).
> > >
> > > If the reviewer can share the concerns more specifically, we are more than welcome to change our manuscript to address the concern. We appreciate the reviewer’s efforts to make our paper more precise.

---

> > > > ### Comment · Reviewer_8gEu · 2022-11-16
> > > > **Reviewer response to #1**
> > > >
> > > > Thanks for these clarifications.
> > > >
> > > > > The foremost message of our paper is to learn a behavior policy $\beta$ with strong generative models to constrain value learning. In fact, the proposed algorithm ARQ does not train a policy with a strong generative model.
> > > >
> > > > I'm confused -- The first sentence seems so say that the paper is about better generative models, while the second seems to say that the proposed algorithm doesn't actually use a better generative model. For what it's worth, I think that a strong paper could be written about either of these topics ((1) better generative models and (2) better algorithms). The concern with the current paper is that it's unclear whether the paper is doing (1) or (2).
> > > >
> > > > > we believe our argument on using strong generative models for the regularization of value function is well-supported (and the next paragraph about relationship with prior methods).
> > > >
> > > > So, is the make takeaway from the paper that "strong generative models are useful, but only if used to regularize the value function"? I agree that the proposed method is novel. What I'm trying to disentangle is _why_ the method works -- is it because of the new algorithm, the strong generative model, or something else?

---

> > > > > ### Author Response · Authors · 2022-11-18
> > > > > **We are getting closer!**
> > > > >
> > > > > > I'm confused -- The first sentence seems so say that the paper is about better generative models, while the second seems to say that the proposed algorithm doesn't actually use a better generative model.
> > > > >
> > > > > The second sentence is regarding the issue of whether the paper is about “a new policy architecture for offline RL.”, the reviewer mentioned in the previous comment, and it seems like there is a small but tricky misunderstanding of the terms “using”, “training”, and “new policy architecture”.
> > > > >
> > > > > We certainly use the score-based generative model, but we did NOT ***train*** a better policy $\pi$ using the score function; we ***use*** the score-based generative model only to approximate $\beta$, and utilize it to learn a regularized Q-function (ARQ) and to indirectly represent a policy (ARQ + $s_\psi$). Therefore, our paper is not about a new policy architecture for $\pi$.
> > > > >
> > > > > > Is the main takeaway from the paper is that “strong generative models are useful, but only if used to regularize the value function"?
> > > > >
> > > > > > For what it's worth, I think that a strong paper could be written about either of these topics ((1) better generative models and (2) better algorithms). The concern with the current paper is that it's unclear whether the paper is doing (1) or (2).
> > > > >
> > > > > Throughout the paper and this rebuttal, we argued that strong generative models are useful – but not only useful – for value regularization. That being said, yes, our main point is using high-fidelity generative models to implement justifiable value function regularization.
> > > > >
> > > > > Given the core message, it is understandable that disentangling our approach into two different topics (generative models vs algorithms) is difficult, because our approach proposes a new algorithm ***via*** a high-fidelity generative model. Certainly, if we only use a strong generative model only for policy representation (similar to the concurrent work [1]), we could examine the effectiveness of recent generative models in the offline RL context. However, that would be a different paper that is loosely coupled to our work where the only commonality is using strong generative models; for instance, there would be a distinct algorithmic difference between ours and [1] as we explained the response to Reviewer 8jQf.

---

### Official Review · Reviewer_8jQf · 2022-10-22

**Confidence:** 4
**Correctness:** 3
**Technical Novelty And Significance:** 3
**Empirical Novelty And Significance:** 3
**Recommendation:** 6

**Clarity, Quality, Novelty And Reproducibility:**

The paper is clearly written. The quality of the paper can be seen in the section above. The novelty of paper is okay though not super original as the technical originality is simply applying a better generative model for the behavior policy. The paper seems to be reproducible as the authors provided code though I haven't run it myself.

**Strength And Weaknesses:**

Strength:

1. The idea of using a powerful generative model for learning explicit behavior policy to mitigate the issue of simply seeking the mode of the behavior actions is reasonable and intuitive.
2. The empirical results are strong and show that the method can outperform prior BC and offline RL methods on D4RL (especially antmaze and kitchen tasks that require stitching) and Robomimic (where the data is mostly demos) tasks.
3. The paper is clearly written and easy to follow.

Weaknesses:

1. I think the main point of using an explicit behavior policy is to avoid seeking the mode, overly mimicking the dataset, and not relying a proxy. However, these points are mostly based on intuition without clear empirical or theoretical evidence/justification. Theorem 1 simply shows the equivalence between soft policy iteration with penalty and policy iteration with KL regularization, which is less relevant to the main point of the paper. Therefore, I think the real justification of why this approach should be preferred is somewhat missing.
2. The empirical results are nice, but I think the authors should compare to [1]. It is unclear if we just need a more expressive policy or we really need to model the behavior policy.

[1] Wang, Zhendong, Jonathan J. Hunt, and Mingyuan Zhou. "Diffusion policies as an expressive policy class for offline reinforcement learning." arXiv preprint arXiv:2208.06193 (2022).

**Summary Of The Paper:**

This paper presents a new offline RL algorithm that directly models the behavior policy with SOTA score-based generative models and use the learned behavior policy as an explicit constraint in offline RL. The authors argue that explicitly modeling behavior policy can avoid failure cases where previous methods with implicit behavior policy constraints seek the mode of the behavior policy and ignore rare but useful actions. Empirically, the methods outperform prior methods in D4RL and Robomimic datasets.

**Summary Of The Review:**

Based on the comments above, I think I would to vote for a weak accept and it would be great if the authors address some of the concerns in the sections above.

---

> ### Author Response · Authors · 2022-11-14
> **To reviewer 8jQf #2**
>
> > The empirical results are nice, but I think the authors should compare to [1]. It is unclear if we just need a more expressive policy or we really need to model the behavior policy.
>
> Thank you for the pointer to the concurrent work. We updated the manuscript and cited the concurrent work that utilizes a strong generative model similar to ours. That being said, our work should be evaluated independently of concurrent works that were not yet published at the time of our submission.
>
> To answer the question, offline RL needs both expressive policy representation and behavior policy modeling. Actually, Wang et al. [1] also say the same in their paper---“we add a term *maximizing action-values* into *the training loss of the conditional diffusion model*, which results in a loss that **seeks optimal actions that are near the behavior policy**”. This can be also confirmed in Equation 3 in their paper in which its  loss term is expressed as a linear combination of BC loss (diffusion loss) and actor loss (Q-optimization loss).
>
> Admittedly, expressive policy representation is an essential piece in both of the works, and we share a very similar motivation that concerns the complexity of behavior policy whose distribution could be multi-modal and/or skewed. However, there is a major difference between the concurrent work and ours. While our implementation is based on solid theoretical ground and performs behavior cloning and value learning in two separate steps, they suggest a method that mixes two objectives via linear combination and optimizes it at the same time. Their approach provides a simple, minimalistic offline RL algorithm as [TD3+BC], but they did not show theoretical support as we do.
>
> Diffusion-RL could contain the same potential pitfall of [TD3+BC] that arises when the two loss functions are just linearly combined, and we can examine the pitfall using the suggested DQP framework. Since the loss function for actor update is $Q - \frac{1}{\alpha} \mathcal{L}_d$, the penalty function of [1] is $p(s,a) = \frac{1}{\alpha} \mathcal{L}-d$. Since the goal of $\mathcal{L}_d$ is to learn the behavior policy $\beta(a|s)$, we can simplify the $\mathcal{L}_d$ as $-\log \beta(a|s)$. To summarize, the penalty function (with some simplification) of diffusion-RL is: $p(s,a) = - \frac{1}{\alpha} \log \beta(a|s)$. Now, we can observe that the penalty function could potentially prefer actions that are more frequently executed by a behavior policy if the hyperparameter $\alpha$ is not properly adjusted. We presume that this is the reason why Wang et al. adapted the $Q$ normalization trick introduced [TD3+BC] in which the penalty function is normalized with the absolute mean value of Q function: $\frac{1}{\alpha} = \frac{ \mathbb{E} \[| Q(s,a) |\] }{\eta}$.
>
> Another potential problem of diffusion RL is its computational cost related to the diffusion steps. Since they directly backpropagate through the reverse diffusion chain, the training can be computationally expensive. They work around this problem by setting a small number of diffusion timesteps, such as N=5, but it could limit the expressivity of the policy. In contrast, our approach does not suffer from such problems because we perform behavior cloning and value learning separately.

---

> ### Author Response · Authors · 2022-11-14
> **To reviewer 8jQf #1**
>
> Thank you for taking the time to review our paper, and we are encouraged that the reviewer finds out that our paper is interesting and easy to follow. We believe the main concern is about the justification of the proposed algorithm ARQ, and we hope our step-by-step justification in the response addresses the concern. If it does, we kindly ask the reviewer to reassess the paper in light of this response. We are happy to answer more if you have any remaining concerns or questions.
>
> > I think the main point of using an explicit behavior policy is to avoid seeking the mode, overly mimicking the dataset, and not relying a proxy. However, these points are mostly based on intuition without clear empirical or theoretical evidence/justification. Theorem 1 simply shows the equivalence between soft policy iteration with penalty and policy iteration with KL regularization, which is less relevant to the main point of the paper. Therefore, I think the real justification of why this approach should be preferred is somewhat missing.
> Thank you for sharing your concern, and helping us to improve our paper. To answer the question, we would like to separate the reviewer’s concern into two parts: (1) justification of the penalty function in connection with the theorem introduced in the paper, and (2) justification of explicitly cloning the behavior policy $\beta$ using strong generative models.
>
> Let’s answer the first part. The equivalence between the penalty function and KL-divergence can be used to theoretically justify the proposed penalty function. That is, the proposed penalty function instantiates the support set constraint that can enjoy the same theoretical property introduced in [BEAR, MBS-QI]. This is because, the induced policy $\pi_p(a|s) := \text{softmax}\big(-p(s,a)\big)$ is a uniform distribution over the supported actions by a behavior policy, and therefore, KL regularization ($KL(\pi || \pi_p)$) only penalizes when $\pi$ tries to select the out-of-support actions. In addition, the KL-regularization with our $\pi_p$ does not prefer any actions that are more common actions that are likely to be executed by a behavior policy, so we can see that the suggested penalty function implements the support set constraint.
>
> The justification for the use of explicitly cloned $\beta$ is straightforward: since the penalty function is defined over $\beta(a|s)$, we directly approximate $\beta$ to implement the penalty function. It is a major difference from other methods, such as BEAR, in which modeling the behavior policy is avoided out of concern for the difficulty of learning the (possibly) complex behavior policy. In contrast, we presumed that the error caused by behavior cloning could be sufficiently small when we utilize recent advancements in the generative modeling community, and the experimental results show that our presumption was indeed correct.
>
> Certainly, the proposed algorithm, Action restricted Q-learning (ARQ), does not directly implement the penalty function; so to speak, there is no Q function training that involves the penalization. However, it is only the result of algorithmic abstraction that eliminates unnecessary computation using our understanding of the consequence of the infinite penalization in the policy iteration framework (for detailed discussion, see 4th paragraph of Section 4.3). Hence, we believe that there is no significant justification gap between the resulting algorithm and the theorem we provided. Please let us know if you still have questions or concerns about this issue.

---

### Official Review · Reviewer_CL7N · 2022-10-25

**Confidence:** 3
**Correctness:** 4
**Technical Novelty And Significance:** 3
**Empirical Novelty And Significance:** 3
**Recommendation:** 8

**Clarity, Quality, Novelty And Reproducibility:**

### Clarity and quality
Both clarity and quality are good.

### Novelty
The use of score-based generative models and the approximated policy iteration procedure in offline RL settings are somewhat novel.

### Reproducibility
The authors provided the code.

**Strength And Weaknesses:**

### Strength
The paper is clearly presented and the proposed method achieve a good performance.

### Weaknesses
I didn't notice any big weakness.

**Summary Of The Paper:**

This paper studies if explicitly modeling the behavior policy is beneficial in offline RL. Focusing on offline RL methods based on value function penalizations, the authors argue that multiple previous methods all design the penalty in a way that reduces the chance of overestimating the value of out-of-support state-action pairs. Based on this observation, the authors hypothesize that a good penalty term should prohibit out-of-support state-action pairs. Based on this penalization scheme, the authors train a score-based generative model to guide an approximated policy iteration algorithm. On standard offline RL benchmark datasets, the authors show that the proposed method could outperform previous methods that do not explicitly model the behavior policy.

**Summary Of The Review:**

I'm inclined to accept this paper because the paper is clearly presented and the proposed method is well-justified.

---

### Official Review · Reviewer_piV5 · 2022-10-27

**Confidence:** 4
**Correctness:** 3
**Technical Novelty And Significance:** 3
**Empirical Novelty And Significance:** 3
**Recommendation:** 6

**Clarity, Quality, Novelty And Reproducibility:**

The paper is well-written and its contributions are clear. One source of confusion seems to be that the authors use \pi_\phi to denote both the learned policy and the behavior policy? Namely, the authors evaluate two variants: one using the score-function and one using \pi_\phi. I interpreted the latter to mean using BC to learn the behavior policy. However, this is confusing because \pi_\phi also denotes the policy learned by the algorithm. It would be great if the authors could clarify this.

Regarding novelty, I think that the comparison to related work can be a bit more thorough, particularly its comparison to other support-constraint algorithms as BEAR, MBS-PI, and IQL. Namely, the latter two seem to propose very similar algorithms. MBS-PI uses policy iteration rather than policy extraction,  and IQL uses a quantile loss rather than sampling from the support. The authors should provide a discussion on why ARQ may be preferred over both.

**Strength And Weaknesses:**

Strengths:

(1) The theoretical unification of existing offline RL algorithms is useful.

(2) The algorithm achieves strong empirical performance on many different tasks, suggesting that modeling the behavior policy is not as difficult as previously anticipated. This opens the door for many new possible offline RL algorithms.

Weaknesses:

(1) The algorithm requires sampling many actions to compute the Bellman backup, which makes it much slower than existing offline RL algorithms. The authors do point this out as a limitation, however.

(2) The algorithm is similar to many existing offline RL algorithms that use a support-constraint. Namely, MBS-PI in [1] looks extremely similar to me. The one difference I noticed is that the former performs policy-iteration, whereas ARQ uses AWR to extract a policy.

[1] https://arxiv.org/pdf/2007.08202.pdf

**Summary Of The Paper:**

The paper proposes a new offline RL algorithm--ARQ--that explicitly models the behavior policy and uses it in a support-constraint in the Bellman backup of the algorithm. The authors make the following contributions:

(1) Relate many existing offline RL algorithms as using different penalty functions in the Bellman backup.

(2) Propose ARQ, that instantiates the penalty function as a support constraint.

(3) Evaluate ARQ and ablations against existing BC and offline RL algorithms.

**Summary Of The Review:**

Though I believe ARQ itself is quite similar to existing ones, and I am not sure why those differences may be advantageous, the algorithm achieves surprisingly good empirical performance. I believe that the paper demonstrates that modeling the behavior policy can be done effectively, and improve upon offline RL algorithms that explicitly avoid this. This, to me, is a surprising and novel finding. Because of this, I recommend that the paper be accepted.

---

> ### Author Response · Authors · 2022-11-14
> **To reviewer piV5**
>
> We deeply appreciate the thoughtful reviews. We are very delighted to find out that most of the main points we want to convey through the paper are well understood by the reviewer. We believe the main concern of the reviewer is regarding its conceptual and algorithmic similarities to prior works, and we hope the following response resolves the concern, and if it does, we kindly ask the reviewer to reassess the paper in light of this response. We are happy to answer more if you have any remaining concerns or questions.
>
> > I think that the comparison to related work can be a bit more thorough, particularly its comparison to other support-constraint algorithms such as BEAR, MBS-PI, and IQL. Namely, the latter two seem to propose very similar algorithms. MBS-PI uses policy iteration rather than policy extraction, and IQL uses a quantile loss rather than sampling from the support. The authors should provide a discussion on why ARQ may be preferred over both.
>
> All four algorithms propose almost the same objective, namely support set constraint, and therefore, every algorithm enjoys similar theoretical properties of the constraint. However, as we reported in the paper, our method shows stronger empirical performance compared to others, which is therefore a practical reason to prefer our method over others. We argue that the strong performance is derived from our instantiation of the constraint, which models the behavior policy $\beta$ with a strong generative model.
>
> Especially, the use of a strong generative model is a key reason why our methods should be preferred over the other methods. This is because a score-based generative model (1) enables the precise support set constraint and (2) allows expressive policy representation. The advantage of a precise support set constraint can be examined when we compare the performance between ARQ and BEAR in which the constraint is implemented indirectly using MMD distance. We could not directly compare MBS-PI since the performance on D4RL datasets has not been reported, but we presume that MBS-PI would fail similarly to BCQ, due to its use of a less powerful generative model (VAE) which would suffer when the $\beta$ is complex or multi-modal. (Note that the authors implemented MBS-***PI*** using a VAE trying to model the behavior policy $\beta$, and therefore, the resulting algorithm MBS-***QL*** is similar to ARQ except for the choice of generative models.)
>
> The advantage derived from an expressive policy representation can be observed by comparing the performance on complex tasks such as adroit, kitchen, and robomimic tasks. These tasks require an expressive policy representation due to their contact-rich nature that makes the optimal policy discontinuous function or distribution. Also, some of the datasets contain human demonstrations, and this could also make policy learning difficult since a policy might have to express a narrow, multimodal distribution defined with a few distinct demonstrations. Since ARQ allows us to leverage the trained BC model that does not suffer from these empirical difficulties, ARQ can avoid potential errors that could occur in policy extraction, and it could result in better performance.
>
> > The paper is well-written and its contributions are clear. One source of confusion seems to be that the authors use $\pi_\phi$ to denote both the learned policy and the behavior policy? Namely, the authors evaluate two variants: one using the score-function and one using $\pi_\phi$. I interpreted the latter to mean using BC to learn the behavior policy. However, this is confusing because $\pi_\phi$ also denotes the policy learned by the algorithm. It would be great if the authors could clarify this.
>
> We apologize for the confusion. Technically, $\pi_\phi$ is not cloning a behavior policy since $\pi_\phi$ is trained via weighted behavior cloning (See the last while loop of Algorithm 1). The effect of the weighted behavior cloning can be easily understood when we assume an extreme case in which $\alpha$ is set with a very large number. In such a case, $\pi_\phi$ will only *selectively* clone actions that have a positive advantage, and therefore, $\pi_\phi$ would perform better than a behavior policy $\beta$. Note that, all across the paper, we use $s_\psi$ to denote the cloned behavior policy.

---

### Decision · Program_Chairs · 2023-01-20

**Decision:**

Reject

**Justification For Why Not Higher Score:**

See meta-review -- The point about explicitly modeling the behavior policy in a BCQ-like fashion is the main focus of the writing of the paper, including the title, though this of course has very limited novelty given that methods like BCQ already propose to explicitly model the behavior policy. The writing seems to somewhat disregard the prior work on explicitly modeling behavior policies, though the revised version of the paper is improved in this respect. The perhaps most interesting new part of the paper is that the method uses a diffusion model when modeling the behavior policy. That said, the paper in its current form does not effectively show that the diffusion model is helpful -- there seem to be experiments that show that it is useful, but these experiments are not discussed and are not the primary focus of the paper's contribution.


**Justification For Why Not Lower Score:**

n/a

**Metareview: Summary, Strengths And Weaknesses:**

This paper introduces a paper that combines ideas from BCQ and AWR into an algorithm, showing strong results. The point about explicitly modeling the behavior policy in a BCQ-like fashion is the main focus of the writing of the paper, including the title, though this of course has very limited novelty given that methods like BCQ already propose to explicitly model the behavior policy. The writing seems to somewhat disregard the prior work on explicitly modeling behavior policies, though the revised version of the paper is improved in this respect.

The perhaps most interesting new part of the paper is that the method uses a diffusion model when modeling the behavior policy. That said, the paper in its current form does not effectively show that the diffusion model is helpful -- there seem to be experiments that show that it is useful, but these experiments are not discussed and are not the primary focus of the paper's contribution.

Overall, the paper is borderline. After discussing the paper with the reviewers and the senior area chair, I am concluding that the issues with presentation and limited novelty generally outweigh the benefits.

**Summary Of Ac-Reviewer Meeting:**

There was no AC-reviewer meeting, but the AC read the reviews in detail, discussed with reviewers, and discussed with the senior AC.